# Aged-Related Changes in Microglia and Neurodegenerative Diseases: Exploring the Connection

**DOI:** 10.3390/biomedicines12081737

**Published:** 2024-08-02

**Authors:** Borrajo Ana

**Affiliations:** Department of Microbiology and Parasitology, Faculty of Pharmacy, Complutense University of Madrid, 28040 Madrid, Spain; anborraj@ucm.es

**Keywords:** microglia, neurodegenetrative diseases, aging, inflammation, oxidative stress

## Abstract

Microglial cells exhibit properties akin to macrophages, thereby enabling them to support and protect the central nervous system environment. Aging induces alterations in microglial polarization, resulting in a shift toward a neurotoxic phenotype characterized by increased expression of pro-inflammatory markers. Dysregulation of microglial cells’ regulatory pathways and interactions with neurons contribute to chronic activation and neurodegeneration. A better understanding of the involvement of microglia in neurodegenerative diseases such as Alzheimer’s and Parkinson’s is a critical topic for studying the role of inflammatory responses in disease progression. Furthermore, the metabolic changes in aged microglia, including the downregulation of oxidative phosphorylation, are discussed in this review. Understanding these mechanisms is crucial for developing better preventive and therapeutic strategies.

## 1. Introduction

Microglia, the immune cells of the central nervous system (CNS), are macrophages originating from yolk-sac progenitors during embryogenesis. These microglial cells exhibit distinct signature genes from other CNS macrophages [1]. Their development follows a stepwise process, indicating differences between prenatal, postnatal, and adult microglia [2]. Initially thought to have hematopoietic origins due to similarities with dendritic cells and peripheral monocytes, recent studies, including fate mapping, confirmed their embryonic yolk sac derivation. While irradiation-induced myeloablation can lead to monocyte infiltration into the CNS parenchyma, it remains essential to distinguish them from true resident microglia [3]. Microglia originate from separate progenitor cells in the yolk sac during embryonic development, contrasting with macrophages [4,5,6,7].

Also, microglial cells are found in the CNS parenchyma, close to neurons, and CNS-associated macrophages (CAMs) and other immune cells, including T and B cells, dendritic cells (DCs), monocytes, natural killer (NK) cells, and NKT cells are found in CNS interfaces, including the meninges (leptomeninges, dura), perivascular space, and choroid plexus [6,7].

Overall, microglia precursors migrate from the yolk sac during embryonic development, proliferate, and acquire their signature genes, contributing to CNS homeostasis through dynamic surveillance and interaction with various components [8]. They colonize the CNS, self-renewing throughout life [8], with minimal contribution from bone-marrow-derived monocytes [9]. These cells play crucial roles, including supporting neurogenesis, pruning synapses, phagocytosing apoptotic neurons, defending against infections, producing and remodeling the extracellular matrix, maintaining myelin health, and removing protein aggregates linked to neurodegenerative diseases [10]. Homeostatic microglia exhibit a highly ramified morphology, responding to stimuli, aging, or CNS pathology, which triggers morphological changes, as noted by Pío del Río-Hortega over a century ago [11].

The use of colony-stimulating factor-1 receptor (CSF1R) inhibitors has provided significant insights into microglial dynamics in the adult CNS. In an interesting work, Elmore et al., in 2014 [12], observed that administering CSF1R inhibitors almost completely eliminated microglial cells in the adult CNS. Following the withdrawal of these inhibitors, microglia rapidly repopulated the CNS. This repopulation was associated with an increase in nestin-positive cells throughout the CNS, suggesting these cells represent microglial progenitors [12]. Initially, it was proposed that the repopulating microglia originated from a neuroectodermal lineage. This hypothesis was based on the observation that microglial cells are myeloid lineage cells and nestin-positive progenitor cells originate from the neuroectodermal lineage. Ref. [13] suggested that repopulated microglia are derived from myeloid progenitor cells. However, recent findings challenge these theories. In a recent study, an adult mouse model was used and it was demonstrated that repopulated microglia do not originate from blood cells, nestin-positive cells, astrocytes, oligodendrocyte precursor cells, or neurons. Instead, after the selective elimination of more than 99% of microglia, the remaining microglia (<1%) proliferated to repopulate the entire brain. Thus, the surviving microglia are the sole source of the new microglia, which rapidly repopulate the brain [13].

In aged mice, the cortex contains fewer microglia compared to young adult mice, and these cells are smaller, less symmetrical, more elongated, and have fewer branches [13]. Aging affects microglia through telomere shortening, DNA damage, and oxidative stress. Telomeres, which are the protective ends of chromosomes, shorten with age, correlating with a decline in microglial self-renewal [14]. Aged microglia express higher levels of tumor necrosis factor-α (TNF-α) and interleukin IL-6, and their dystrophic morphology hinders spatial learning [13]. Senescent microglia show increased levels of pro-inflammatory cytokines, reduced levels of chemokines, and decreased phagocytosis of amyloid beta (Aβ) fibrils [13]. They also exhibit a ‘primed’ phenotype, characterized by an exaggerated and uncontrolled inflammatory response to immune stimuli [13].

The intricate role of factors such as CSF1 and its receptor CSF1R in microglia development and maintenance highlights the complexity of their regulation [15]. The absence of yolk-sac macrophages and the failure of microglia colonization in CSF1R-mutant mice emphasize the importance of this signaling pathway [16]. Additionally, the significant reduction of microglia in response to CSF1R inhibition further underscores the critical role of these factors in microglial population dynamics. The findings from microglia-specific Csf1r knockout mice and interleukin-34 ablation in neuronal progenitors emphasize the specific and dose-dependent nature of factors influencing microglia [17]. The involvement of transcription factors, interferon regulatory factor 8 (IRF8), and PU.1, along with the anti-inflammatory cytokine transforming growth factor-β (TGF-β), further underscores the complexity of regulatory mechanisms governing microglial development and homeostatic functions [3]. Homeostasis is defined as a relative constancy of a set point formed in certain conditions, and maintaining homeostatic microglial function demonstrates an effort to restore the deviating set point due to aging in the CNS environment [3].

In the early nineties, there was evidence that indicated that aged microglia had greater phagocytosis activity [18]. Different studies have corroborated that aged microglia have a primed phenotype [19] and a tendency to react more intensely to stimulation arises, attributed to their downregulation of homeostasis and inhibitory genes and increased expression of activating ones [5,20]. These modifications have been documented in various areas of the aged brain, particularly notable in white matter-resident microglia, indicating their responsiveness to age-related changes in myelin [5].

Aging microglia respond to an inflammatory stimulus can induce heightened levels of pro-inflammatory cytokines such as IL-1β and IL-6 or TNF-α [21]. Nevertheless, research indicates that while aged microglia still express IL-10 and IL4R-α at levels akin to young microglia, these interleukins fail to mitigate inflammation due to diminished cellular responsiveness [22]. Consequently, aging microglia fail to restore homeostasis, exacerbating cognitive impairment, sickness, and depressive-like behaviors to levels typically observed in neuroinflammation scenarios [5,22].

An important microglial function is to phagocytose synapses, an essential process during development in order to ensure correct neuronal connectivity [22]. However, this occurs as a mechanism, mediated by microglia, to ensure neuronal plasticity and depends on neuronal activity [5,23]. It is still unknown whether microglia-mediated synaptic phagocytosis is responsible for the decline in synaptic loss that correlates with cognitive decline, as observed in aging human brains [24].

The present review article summarizes our current knowledge of the functional and phenotypic properties of aging microglia, with a special focus on the contribution of aged microglia to the development and progression of neurodegenerative diseases such as Alzheimer’s Disease (AD) and Parkinson’s disease (PD).

## 2. Microglial Cells Morphologies and Different Microglia Shapes in CNS Regions

Microglial cells, under normal conditions in a healthy brain, exhibit a ramified morphology with numerous long, thin, and highly branched processes [25]. Initially, it was believed that a ‘resting state’ was a typical feature of ramified microglia until advancements in molecular techniques, such as in vivo two-photon imaging, uncovered their highly mobile protrusions capable of actively surveying, recognizing, and reacting to adverse environmental conditions [26].

It was indicated that microglia with branched structures predominantly express genes linked to stable brain functions such as synaptic integrity, neuronal development, and overall cell balance by transcriptomic studies [27]. Branched processes facilitate continuous interaction with neurons and other glial cells, either directly or through secreted signaling molecules [28]. Upon sensing changes in the environment, microglial cells swiftly migrate toward the stimuli, aided by chemotactic signals [26]. The activation of microglial cells involves a morphological shift from a ramified to an amoeboid state, characterized by an enlarged cell body, shorter processes, and numerous cytoplasmic vacuoles [28]. This morphological transformation is accompanied by functional changes such as antigen presentation, phagocytosis, and migration [26]. 

Other phenotypes of microglial cells include a bipolar/rod-shaped morphology, serving as a transitional state between ramified and amoeboid states [29] and exhibiting distinct transcriptome profiles and heightened proliferative and phagocytic abilities [29]. These activated microglia cells are commonly found in the aged brain and in neuropathological conditions, where they often surround injured axons, assuming neuroprotective roles and facilitating processes such as synaptic remodeling [29]. Various other microglial morphologies have also been identified, including hypertrophic microglia, senescent microglia, satellite microglia, Gitter cell-like microglia, and dark microglial cells [30].

In the presence of pathology, microglial cells change both their shape and function, undergoing a transformation from ramified to hypertrophic and ameboid phenotypes, which is a progression that continues to guide research today [31]. The build-up of senescent microglia, along with their impaired immune functions and interactions with other brain cells, could play a crucial role in the development of neurodegenerative diseases [31]. The unique characteristics of yolk sac-derived microglial cells, which have a limited capacity for repopulation, suggest an intrinsic link to their senescence, potentially contributing significantly to age-related neurodegenerative diseases [31]. 

Regarding the satellite microglia, this subgroup is now identified by its distinctive association with neuronal cell bodies [32]. The portion of the axon and the satellite microglial cells overlap and action potentials are initiated [32]. It has been observed both during development and adulthood in mice, with a preferential association with excitatory neurons [32], and additionally, this subpopulation has been noted in the cerebral cortex of adult rats and adult non-human primates using non-invasive two-photon in vivo imaging [33], indicating conservation across species. These cells show soma migration, interacting with both Purkinje neuronal cell bodies and proximal dendrites, are less ramified than their cortical counterparts, and exhibit reduced surveillance. This finding suggests that satellite microglia are heterogeneous in their dynamics under steady-state conditions [33]. Beyond the distinct morphological properties of microglia revealed by light or fluorescence microscopy, high-resolution electron microscopy has uncovered the presence of microglia filled with cellular debris, similar to the fat granule or Gitter cells initially described by Río-Hortega in 1920 during aging, age-related sensory loss, and Werner syndrome in mice [34]. Recently, electron microscopy also identified a unique microglial population, the “dark” microglia, in the adult and aged mouse hippocampus (CA1 region and dentate gyrus), cerebral cortex, amygdala, and hypothalamus. These cells coexist with the typical microglia but display several ultrastructural features, including their size, morphology, long stretches of endoplasmic reticulum, interactions with neurons and synapses, and association with the extracellular space. Unlike typical microglia, they exhibit markers of oxidative stress, such as a condensed, electron-dense cytoplasm and nucleoplasm, giving them a dark appearance in EM, along with Golgi apparatus/endoplasmic reticulum dilation, mitochondrial alterations, and a partial to complete loss of heterochromatin pattern [34]. In neurons, changes to the heterochromatin pattern are associated with cellular stress, aging, and brain disorders such as schizophrenia and Alzheimer’s disease [35].

Grabert et al., in a brilliant work, studied how the brain microglia in the hippocampus decreased the expression of genes related to environment sensing with a profile between pro- and anti-inflammatory states, producing more vulnerability to aging and disease-related protein deposition [36]. Particular features of the hippocampus that might influence microglial cell activity are microglial gray matter content and density [37]. However, the immunophenotypic variation of striatal and cortical microglial cells was similar. The microglia, in healthy conditions, in the hippocampus, presented a higher “immune-vigilant” phenotype which can be related to a higher chronic inflammatory microglial response in AD pathology to plaque formation typical of this disease. The fractalkine receptor, CXCR3, has been shown to be involved in neuron-microglia communication, so hippocampal lower values of CXCR3 might provoke less microglial cells control [37]. These changes may influence microglial cell activity patterns and predispose this activity toward hippocampal-related neurocognitive diseases, where neuroinflammation and microglial cell activation and neuroinflammation play crucial roles [37].

Temporal lobes including motor areas and the prefrontal cortex are extensively connected, playing important roles in top–down behavioral control and language processing, and exhibit the highest degree of age-related atrophy, in the case of the prefrontal cortex, with a significant decline in prefrontal gray matter volume compared to other areas such as superior parietal cortices and the inferior temporal. In the healthy cortex, microglia/macrophage-specific inflammatory receptor CD11b expression on microglia is consistently lower than in the spinal cord, and the microglia protein, CD40, expression is lower in the cortex compared to the cerebellum. Conversely, CXCR3 expression is higher in the cortex than in the cerebellum [37].

Also, microglia constitute 12% of the cellular population in the substantia nigra, a particularly dense population [37]. CXCR3 is more abundant in the striatum than in other structures such as the cerebellum [37]. Another important receptor in microglia–neuron interactions is the membrane glycoprotein CD200, which is downregulated with age in the substantia nigra [37]. A high density of microglial cells reduces CD200 receptor expression, and a decreased dopaminergic population can diminish the regulatory influence of neurons on microglial cells. Such an environment may lead to a reduction in the production of trophic factors or an increase in inflammatory molecules, which can be detrimental over time.

## 3. Characteristics and Phenotypes of Aged Microglia

It has been shown in studies carried out in healthy rodents that microglia constitute between 5% and 12% of all specific cells in the CNS. However, the distribution is diverse and some brain areas display higher densities of microglia [38]. Interestingly, the nigrostriatal system shows higher microglia densities compared to adjacent brain regions, comprising the substantia nigra (SN) and the caudate putamen (CPu) [39]. In humans, microglia represent 0.5–16.6% of all brain parenchymal cells, with higher numbers observed in white matter than in gray matter [40]. 

Different works have tried to address age-related changes in microglial cell numbers in diverse species with different results. While there have been no obvious changes in the number of Iba1+ microglia in the hippocampus of aged rats [41], reductions in the number of microglia were detected in the aged nigrostriatal system and cerebral cortex [40]. Conversely, in aged rhesus monkeys (25 to 35 years old), there was an increase in the number of microglia, accompanied by heterogeneous intracellular inclusions suggesting heightened phagocytic activity but reduced particle digestion ability [18]. Aging human microglia display dystrophic morphologies, characterized by residual process fragmentation, reduced branching, deramified dendritic arbors, and cytoplasmic beading, varying across brain regions [42]. Dystrophic microglia are differentiated at the morphological and functional levels from dark microglia, with highly branched morphology, which shows condensation of their cytoplasm and nucleoplasm, accompanied by contraction of the cytoplasm, Golgi apparatus, and dilation of the endoplasmic reticulum [43].

Homeostatic microglial functions decrease with aging, along with these morphological changes. Maintaining homeostatic function involves timely and appropriate responses at each life stage. Excessive or insufficient microglial responses can hinder tissue repair after CNS damage, underscoring the importance of tightly regulating the transition from steady-state homeostasis to an immunomodulatory mode during pathological conditions. Microglial immune checkpoints, mechanisms that prevent uncontrolled responses, have been proposed [44]. CX3CR1, expressed on dendritic cells, monocytes, and microglial cells, plays a crucial role due to its nature as a transmembrane protein and chemokine for leukocyte migration [45]. Its ligand, CX3CL1, expressed on neurons, regulates microglial functional phenotype and prevents hyperactivation under inflammatory conditions. For instance, mice lacking CX3CR1 showed exaggerated neurotoxic microglial responses and increased neuronal death when exposed to lipopolysaccharide (LPS) stimuli in the CNS. Additionally, CD200 receptor (CD200R) interaction with its ligand on neighboring cells, including neurons, astrocytes, oligodendrocytes, and endothelial cells, attenuates microglial activation, particularly in inflammatory contexts [3].

Some works have analyzed the changes in microglial polarization responses associated with age. In 2019, Wang et al. compared microglial markers in 2-, 6-, 18-, and 28-month-old rat brains. Markers associated with different microglial phenotypes were examined, including M1 markers such as IL-1β and TNF-α, M2 markers such as IL-10 and arginase 1 (Arg1), A1 markers such as lipocalin-2 (Lcn2) and complement C3 (C3), and A2 markers such as brain-derived neurotrophic factor (BDNF) and glial-cell-line-derived neurotrophic factor (GDNF). Additionally, M1 markers increased and M2 markers decreased in aged rats [46]. Also, NF-κB can be considered as a crucial factor in the polarization of microglia, promoting the M1 microglia phenotype and being inhibited by regulators switching microglia from M1 to M2. NF-κB inactive is present in the cytoplasm without being stimulated to bind to IκB/NF-κB [46].

In aged rats, A2 markers (BDNF and GDNF) decreased while A1 markers (Lcn2 and C3) increased, suggesting an alteration in midbrain glial cell phenotypic polarization, potentially contributing to neurodegenerative diseases such as PD due to age-induced DA neuron loss (Figure 1).

The idea of microglia polarization is currently debated because the M1/M2 paradigm may oversimplify in vivo activation. The heterogeneity of the microglia phenotype related to various pathogenic situations can be further defined using transcriptomic and proteomic analyses.

In aged mice, a loss of the M2-like phenotype has been demonstrated, evidenced by suppressed anti-inflammatory IL-4/IL-13 signaling [46]. There is also an observed increase in the upregulation of TLRs, activation markers (MHCII, CD68, and CD86), and CD11b in aging brains across various species [47], indicating a predominant M1-like phenotype associated with neurotoxic responses (Figure 1). 

One mechanism to demonstrate the presence of reactive microglia in the aging brain is the age-associated loss of endogenous microglia regulatory pathways. For example, the signaling pathway that promotes the inactivity of microglial cells in which TGFβ is involved is related to a reduction in the protective function of microglia [48]. 

Furthermore, notable downregulation of microglial receptors involved in microglia-neuron interactions, such as the purinergic P2Y receptor 12 (P2Y12R), which is crucial for regulating microglial activation and phenotypic transformation, is evident in the aging brain [49]. Additionally, age-related neurodegeneration leads to the depletion of neuron-derived CX3CL1 ligand, which normally maintains microglial cells in a quiescent state [26]. These alterations in endogenous regulatory factors contribute to microglial dystrophy, neurodegeneration, and chronic activation [26].

Transcriptomic studies have helped advance the study of global changes in gene expression associated with aging. Soreq et al. (2017) analyzed the transcriptomic profiles of different types of microglia and neuronal cells in different regions of the human brain in 480 subjects of widely varied ages (between 16 and 106 years), and an increase in cell-specific genes was observed. Microglial cells in all brain regions strongly predict biological aging [50]. In other studies, signature genes associated with normal aging were identified, including high expression of genes encoding TNF family ligands, vesicle release proteins, and proinflammatory cytokine high-mobility group box 1 (HMGB1) [51]. 

HMGB1 mediates microglia priming in the aging brain and desensitizes aged microglia to an inflammatory insult through its inhibition [52]. Holtman et al. (2015) conducted a co-expression analysis, revealing changes in primed microglia from healthy, aged animals and models of accelerated aging and neurodegeneration. They identified a common expression profile characterized by the upregulation of immune receptors, phagosome- and lysosome-related genes, and lipoprotein Apoe [53]. This signature differed from the acute inflammatory gene set induced in animals treated with lipopolysaccharide (LPS), where genes related to the NF-Κb pathway and Toll-like and NOD-like receptor (TLRs and NLRs) signaling were highly enriched. Microglia cells characterized by increased mammalian target of rapamycin (mTOR) signaling had higher transcript levels of ribosomal genes and interferon (IFN) α/β signaling [53]. 

In Zhang and colleagues’ 2021 research, they compared typical elderly mice, Ercc1ko-knockout mice lacking DNA repair mechanisms, and transgenic models of Alzheimer’s disease and Amyotrophic Lateral Sclerosis. Their study revealed shared microglial inflammatory gene networks between normal aging and pathological conditions, which contribute to age-related microglial priming. Unlike the acute inflammatory gene networks induced by LPS, such as NFκB signaling activation, primed microglia in these aging models exhibited increased expression of MHCII and other genes encoding CD11c integrins and CXC-chemokine receptor 4 on cell surfaces. Gene expression analysis demonstrated that Ercc1 deletion in microglia led to a temporary aging signature, distinct from a priming or disease-related microglia gene expression profile. 

One of the main functions of microglial cells is synapse pruning through a phagocytic elimination process in a healthy developing brain. This is mediated through a mechanism that involves the recognition of complement proteins at synapses via microglial receptors [26]. Complement-mediated clearance mechanisms are known to be involved in the aging process although not observed in healthy adult microglia, leading to an altered microglial phenotype that may contribute to neurodegeneration [26]. 

In the elderly brain, accumulated on synapses, CIq and C3 proteins are present, activating phagocytic complement receptors C1qR and C3R expressed in microglia, initiating the removal of healthy synapses [54]. These findings suggest a harmful role for complement-mediated phagocytosis in aging, contributing to neurodegeneration. 

Through the classical or alternative pathways, the complement system can be activated. The classical pathway is initiated when C1q, C1r, and C1s of the C1 complex bind to apoptotic cells, hyperphosphorylated tau, or antigen–antibody complexes via C1q. The lectin pathway is triggered when mannan-binding lectin (MBL) forms a complex with MASP1 and MASP2, binding to microbial carbohydrates. C3 convertase is formed by cleaving C4 and C2 to generate C4b2b in both pathways [55]. C3 is cleaved to produce C3a, promoting chemotaxis and microglial activation via C3aR, while C3b can be cleaved to iC3b to facilitate opsonization. C5 is cleaved into C5a, inducing a potent inflammatory response and acting through C5aR1 to enhance chemotaxis and glial activation, and C5b binds to C6, C7, C8, and C9 to form the membrane attack complex, leading to lysis. The alternative pathway can be activated by spontaneous hydrolysis of C3 to C3-H2O, enabling factors B and D to generate C3 convertase (C3(H2O)Bb), which then cleaves other C3 molecules into C3b. This pathway forms an amplification loop, where factor B binds to C3b and is cleaved by factor D to produce C3bBb, continuing the cleavage of C3 to enhance activation [54] (Figure 2).

Age-related functional change in the brain is also related to dysregulation of signaling of Ca^2+^ [46], which can produce a variety of neuropathological conditions, including the dysregulation of Ca^2+^ channels, dysfunction in mitochondria, and the endoplasmic reticulum impeding Ca^2+^ homeostasis in the aging brain [55]. Changes in intracellular Ca^2+^ levels are produced and microglia express various plasma membrane ionotropic and metabotropic receptors, P2X receptors, and P2Y receptors (nucleotide receptors) [55].

Generally, defects in neuronal autophagy regulation, related to Ca^2+^ regulation in the ER and mitochondria, contribute to several diseases, such as AD and PD [55].

Studies have revealed early Ca^2+^ dysregulations in neurons and glial cells. This situation can promote good astrocyte–neuron communication in the CNS, signaling increase Aβ deposits and disrupting gliotransmitter release. When Ca^2+^ dysregulation takes place first in the glia and then propagates to neurons, targeting the glia early can be a possible therapeutic option for AD [55]. ER, mitochondria, and lysosomes mediate intracellular Ca^2+^ homeostasis and establish a mechanism to avoid proteinopathy, ROS production, oxidative stress, and neuroinflammation, situations that are produced by intracellular Ca^2+^ dyshomeostasis [55]. Additional signaling pathways that play a crucial role in regulating various key functions of microglia, such as phagocytosis and inflammatory responses, also activate these receptors [26]. Live visualization of microglial Ca^2+^ dynamics is being utilized to investigate functional and structural changes in the intact brain [56]. In a recent and interesting study by Olmedillas del Moral and colleagues, in vivo experiments were carried out in young adult, middle-aged, and old mice to evaluate intracellular Ca^2+^ signaling and to perform a process extension analysis of cortical microglial cells. They characterized a complex and nonlinear relationship between the properties of intracellular Ca^2+^ signals and the age of the animals. The ATP process in old mice (18–21-month-old) showed faster but more disorganized movement compared to young adult mice (2–4-month-old). These findings highlight two distinct phenotypes of aging microglia: a reactive phenotype that increases with age, and a bell-shaped relationship between the frequencies and durations of spontaneous Ca^2+^ transients in middle-aged mice. 

Other phenotypes showed that an ATP source moved faster and in a more disorganized manner in old mice. Ca^2+^ dysregulation is also prominent in response to injury, inflammation, and neurodegenerative diseases [56]. In another study by Brawek and colleagues, a microglia-specific microRNA-9-regulated viral vector was employed to express a genetically-encoded ratiometric Ca^2+^ sensor, Twitch-2B, in microglial cells. In intact in vivo microglia, steady-state intracellular Ca^2+^ levels were found to be very homogeneous and low. However, these levels increased significantly after acute slice preparation and cell culturing, accompanied by an upregulation in the expression of activation markers CD68 and IL-1β. These findings highlight the steady-state intracellular Ca^2+^ level as a versatile marker for microglial activation, which is highly responsive to the cellular environment [57].

Distinct phenotypes have been identified based on sex and brain regions. An age-related increase in senescence-associated genes and higher expression of cyclin-dependent kinase inhibitors, such as Cdkn2a/p16^INK4^, were specifically noted in the hippocampus of older female mice [58]. The highest occurrence of p16-positive brain cells was found in monocytes and microglial cell populations. Transcriptomic profiling of these two myeloid clusters revealed an upregulation of chemoattractant SASP factors such as secreted phosphoprotein 1 (Spp1), MHC genes, lysosomal stress markers (Lgals3 and Lgals3bp), and senescence-related genes (Cdkn2a, Cdkn1a, and B cell Lymphoma (Blc2)). Although sex influenced diverse age-related signatures in the aging hippocampus, different works indicated that systemic clearance of p16-positive cells improved cognitive functions in neurodegenerative conditions [5]. Consistent with previous studies, spatially resolved transcriptomic analysis in brain sections of aged animals confirmed the age-related increase of Cdkn2a expression. Furthermore, the microdomains containing senescent cells were heterogeneously distributed in the cortical gray matter, hippocampus, and white matter and were mainly represented by clusters of cells expressing endothelial cells, markers of microglial cells, oligodendrocytes progenitor cells and oligodendrocytes [59].

In general, the data reviewed evidenced that age-associated damage of microglial responses throughout life contributes to the development of neurodegenerative diseases.

## 4. Aged-Microglia and Neurodegeneration

Aged microglia exhibit both damage- and neuro-protective capacity, such as from low and sustained secretions of molecules that drive inflammation often observed in neurodegenerative diseases [60]. Microglial cells with dystrophia have been observed in both aging brains and the brains of patients with neurodegenerative diseases. They are commonly found near areas of tau pathology and amyloid plaques in AD brains, as well as near Lewy bodies in brains affected by dementia with Lewy bodies [61]. 

Aging microglia in models with a β-amyloid burden, exhibiting reduced phagocytic capability, were also noted to express the cytokines TNF-α and IL-1β, unlike microglial cells that successfully phagocytosed β-amyloid [62]. The inflammatory cytokines expression is believed to exacerbate microglial dysfunction and perpetuate AD pathology. Additionally, microglial cells have demonstrated the ability to internalize and degrade extracellular α-synuclein aggregates in cell culture studies, a process apparently regulated by the activation state of the cells, as activation with LPS reduced this activity [62]. 

The role of iron-rich microglia in neurodegenerative diseases is significant. Recent studies have identified ferritin-positive dystrophic microglial cells associated with amyloid plaques and neurofibrillary tangles [62]. Iron accumulation has also been observed in the hippocampus and, notably, in the amyloid plaques of AD patients. The interaction between iron and other metal ions contributes to the toxicity of β-amyloid oligomers [62]. Excessive iron accumulation occurs in regions affected by Parkinson’s disease (PD), such as the substantia nigra, and is associated with Lewy bodies, where dystrophic ferritin-positive microglia have been analyzed [63].

Greater secretion of pro-inflammatory cytokines in microglial cells in vitro has been related to disruption in iron homeostasis [62]. Iron exposure has been related to a higher risk of developing PD [64], and short-term exposure to iron has been shown to enhance neurotoxicity in rat primary microglia cells [62]. Neuromelanin, a protein that stores iron in neuron cells, can be engulfed by microglia cells attracted to degenerating neurons, leading to an increase in the iron content of these cells. Rathnasamy et al., in 2013, determined that neuromelanin phagocytosis can induce an increased release of pro-inflammatory cytokines and reactive oxygen species (ROS), thus driving inflammatory processes that can contribute to neuronal degeneration further [65]. 

It is conceivable that the accumulation of iron in microglial cells could initially serve as a protective mechanism against iron toxicity in the brain. However, this accumulation may eventually impair the microglia themselves, contributing to the accelerated aging signature observed in neuronal degeneration [62].

## 5. Metabolism and Oxidative Stress in Aged Microglia

A finely tuned energy metabolism is crucial for immune cells to maintain balance and fulfill the requirements of bio precursors essential for mounting an effective immune response [66]. However, aging produces the decay of nutrient-sensing networks and a decline in mitochondrial efficiency and integrity [66]. Notably, aged microglia also exhibit altered metabolism, with a downregulation of genes involved in oxidative phosphorylation [66]. This regulation could be a response to changes in energy metabolism or due to an imbalance of growth factors in the brain environment, leading to heightened levels of mTOR signaling in aging microglia [21]. mTOR plays a pivotal role in regulating cell growth and metabolism across all cell types, with its inhibition associated with increased longevity [66]. In aged microglia, augmented mTOR signaling adds another layer of regulation to the primed phenotype by boosting mRNA translation [21].

Activation of the PI3K-AKT-mTOR signaling pathway in microglia is triggered by growth factors and cytokines [21]. AKT activation drives the sequestration of glucose into glycogen, resulting in energy depletion and succinate accumulation [67]. Succinate, stabilized by mTOR signaling, promotes the transcription of pro-inflammatory genes via hypoxia-inducible factor 1α (HIF1α) [67]. Concurrently, mTOR complex 1 (mTORC1) stimulates ribosomal gene transcription through various mechanisms [5], culminating in heightened translation of pro-inflammatory factors such as TNF, IL-1β, and IL-6 in aged microglia [21]. Transcription factors such as PPARγ facilitate the formation of lipid droplets in myeloid cells, suggesting a similar mechanism in microglia, downstream of mTOR [5]. COX2, located on the droplet membrane, enhances PGE2 production, creating a positive feedback loop [5]. The PI3K-AKT-mTOR pathway, crucial for microglial response to amyloid-β (Aβ) deposition, peptide is considered a critical neurotoxic agent in AD pathology and is activated by TREM2 [67] (Table 1). Reduced expression of TREM2 variants is associated with neurodegeneration [67].

Antignano and co-researchers recently unveiled that aged microglia heighten mTORC1 signaling and downstream mRNA translation via the 4EPB1-EIF4E axis [21]. The heightened mTOR-dependent phosphorylation of 4EBP1 leads to increased translation and subsequently elevated protein levels of inflammatory receptors and cytokines in aged microglial cells compared to their younger counterparts, an effect that could be mitigated by the loss of Rheb1, the upstream positive regulator of mTORC1. In an interesting study, Holtman et al. reported that microglial cells with heightened mTOR signaling exhibit higher transcript levels of ribosomal genes [21,53]. 

mTORC1 signaling has been also identified as a promising target for the treatment of major depressive disorder. Luo et al. explored whether the mTORC1 signaling pathway is involved in synapse loss in the hippocampus caused by chronic stress. After successfully establishing the chronic restraint stress model, significant changes in the mRNA levels of certain immediate early genes were observed, indicating neuronal activation and changes in protein synthesis [76]. There was a notable downregulation of glutamate receptors and postsynaptic density protein 95 at both the mRNA and protein levels. Synaptic fractionation assays indicated that chronic stress resulted in synapse loss in both the ventral and dorsal hippocampus. These effects were related to the mTORC1 signaling pathway, as protein synthesis and phosphorylation of downstream signaling targets were reduced following chronic stress. Moreover, intracerebroventricular infusion of rapamycin induced depression-like behaviors and inhibited the antidepressant effects of fluoxetine [76]. This study demonstrates that the mTORC1 signaling pathway plays a crucial role in mediating synapse loss provoked by chronic stress in diseases such as AD and PD and contributes to the behavioral effects of antidepressant treatment.

Metabolic shifts can directly produce mild inflammation in aged microglial cells and upregulation of pro-inflammatory prostaglandin E2 (PGE2) signaling [77]. The EP2 receptor permits PGE2 to act through this receptor, promoting the sequestration of glucose into glycogen via the AKT–GSK3β–GYS1 pathway, resulting in decreased mitochondrial respiration, production of ATP, and reduced levels of glucose. The decreased de novo NAD + synthesis is produced by this energy-depleted state, damaging the NAD-dependent Sirt3-mediated deacetylation of mitochondrial complex II subunits, thereby reducing succinate dehydrogenase activity [78]. Consequently, the accumulation of a TCA cycle metabolite, succinate, stabilizes the activity of hypoxia-inducible factor 1α, which activates pro-inflammatory cytokines [77]. The inhibition of PGE2, either pharmacologically or genetically, decreases the expression of pro-inflammatory cytokines in the blood and hippocampus of aging animals, enhances hippocampal plasticity and memory function, and reverses this phenotype towards a more anti-inflammatory profile in peripheral macrophages and microglial cells [77]. Importantly, the peripheral block of the EP2 receptor with a brain-impermeant EP2 antagonist in aging animals yields similar outcomes, suggesting that age-associated brain inflammation and cognitive decline are not irreversible processes and can be altered by intervening in peripheral cells [21].

Microglial lipid metabolism undergoes alterations with aging, evident through the accumulation of cytoplasmic inclusions such as lipid droplets [21]. Marschallinger et al. elucidated the functional implications of this aged phenotype. Analysis of the transcriptome in Lipid Droplet-Associated Microglia (LDAM) showed a significant deficiency in phagocytosis and an elevation in ROS and inflammatory cytokine production. Analyses of the RNA sequence of LDAM uncovered a transcriptional profile predominantly driven by innate inflammation, differing from previously documented states of microglial cells. An impartial CRISPR-Cas9 screen pinpointed genetic modifiers of lipid droplet formation; surprisingly, variants of several of these genes are implicated in autosomal-dominant forms of human neurocognitive disorders. Notably, LDAM exhibited a gene signature partially overlapping with that of microglia from LPS-treated mice [78]. In this study, it has been proposed that LDAM contributes to age-related and genetic forms of neurodegeneration [78].

## 6. Studies of Aged Microglia in Alzheimer’s and Parkinson’s Disease

In mouse models of AD, microglia seem to construct a barrier that diminishes the neurotoxic effects of protofibrillar Aβ in amyloid plaques [79]. Employing high-resolution confocal and in vivo two-photon imaging in AD mouse models, Condello et al. demonstrated that this microglial barrier inhibits outward plaque expansion, leading to compact plaque microregions with low Aβ42 affinity. Regions devoid of microglial cells display less compactness but possess high Aβ42 affinity, fostering the development of protofibrillar Aβ42 hotspots linked to severe axonal dystrophy. With aging, microglia coverage diminishes, resulting in enlarged protofibrillar Aβ42 hotspots and more severe neuritic dystrophy [79]. Anti-Aβ immunotherapy or the deletion of the CX3CR1 gene promotes expansion of microglial cell coverage and reduces dystrophy of the neurites [79]. The breakdown of the microglial barrier and the accumulation of neurotoxic protofibrillar Aβ hotspot accumulation represent potential targets for therapy and clinical imaging in AD. 

Recent research highlights the potential of young microglia in restoring amyloid clearance in aged microglia, as demonstrated in ex vivo brain slice co-cultures by Daria et al. [80]. To explore the role of microglia in phagocytosis of amyloid plaque, a novel ex vivo model was devised, co-culturing organotypic brain slices from amyloid-bearing AD mouse models with young, neonatal wild-type mice. Curiously, co-culturing induced the recruitment, proliferation, and clustering of old microglia around amyloid plaques, resulting in the clearance of the plaque halo. There is a synergistic effect of old or young microglia due to the depletion of either old or young microglial cells that impeded amyloid plaque clearance. Old microglial cells have been exposed to conditioned media from young microglial cells or the addition of granulocyte-macrophage colony-stimulating factor that induced reduced amyloid plaque size and microglial cell proliferation [80]. There is a reversal of mitochondrial dysfunction in AD that is suggested by these data and their ability to phagocytize can be modulated to restrict the accumulation of amyloid. This innovative ex vivo model provides an important platform for identifying, screening, and testing compounds aimed at enhancing microglial phagocytosis therapeutically.

In transgenic AD mice, exacerbated age-dependent microglial activation and disturbances in cytoskeletal regulations contribute to further neurodegeneration [81]. This research provides a comprehensive analysis of the gene expression patterns in the APP23 model for AD and control mice, exploring the impact of aging on these patterns. These results are linked to different symptomatic and pathological features of the model such as changes in soluble Aβ levels [81]. A distinct two-phase expression profile was observed, with the first phase resembling features found in young carriers of familial AD mutations, while the second phase mirrors the progression of human AD pathology. There is a noticeable increase in microglial activation and lysosomal pathways during this later phase, alongside a decrease in neuron differentiation and axon guidance pathways. Curiously, these alterations are associated with aging but are more pronounced in APP23 mice [81].

Moreover, complement factor C3 secreted by reactive astrocytes interacts with microglial C3a receptor (C3aR), mediating Aβ pathology and neuroinflammation in AD mouse models [82]. 

In this study, it is reported that astrocytic complement activation also regulates Aβ dynamics in vitro and amyloid pathology in AD mouse models through microglial C3aR [82]. In primary microglial cultures, it has been demonstrated that acute activation of C3 or C3a promotes microglial phagocytosis, while chronic treatment with C3/C3a diminishes it. The impact of chronic exposure to C3 can be mitigated by co-treatment with a C3aR antagonist and by genetically deleting C3aR. Furthermore, it is demonstrated that neuroinflammation and Aβ pathology worsen in transgenic mice due to astroglial NF-κB hyperactivation and resulting C3 elevation [82]. Additionally, therapy with a C3aR antagonist reduces microgliosis and plaque load, indicating a complement-dependent intercellular communication between Aβ and activated astroglial NF-κB, triggering the extracellular release of C3 that interacts with neuronal and microglial C3aR to impede Aβ phagocytosis and modify cognitive function [82]. This feedback loop can be disrupted by C3aR inhibition, suggesting therapeutic potential in chronic neuroinflammation conditions. 

However, the lack of C3aR in APP transgenic mice results in decreased Aβ deposition, suggesting a complex role of microglia in AD pathogenesis [83]. Additionally, the regulation of the clearance of soluble Aβ independently of phagocytosis is a new duty for the microglial complement receptor 3 (CR3). Remarkably, this leads to cultured microglia lacking CR3 and exhibiting greater efficiency in degrading extracellular Aβ. Moreover, a small molecule modulator of CR3 reduces extracellular soluble Aβ levels and Aβ half-life in brain interstitial fluid, indicating a potential new therapeutic target in AD [83].

Interestingly, studies utilizing microglia ablation in APP transgenic mouse models revealed that neither Aβ plaque formation nor amyloid-associated neuron dystrophy depended on the presence of microglia [84]. Further investigations are crucial to comprehensively understand how microglia contribute to AD onset and progression, with particular emphasis on aged microglia to elucidate the impact of aging on microglial functions in AD. 

In PD, pronounced microglial reactions have been observed in the human substantia nigra (SN) associated with extraneuronal neuromelanin deposits [39]. Human neuromelanin can induce microglia-mediated neuroinflammation and neurodegeneration in rodents [39], suggesting its role as a potent trigger for microglial activation. Rodent-toxin-based PD models, such as 1-methyl-4-phenyl-1,2,3,6-tetrahydropyridine (MPTP), have elucidated the contribution of microglia to mDA neuron degeneration [84]. Machado and col. conducted analyses of transgenic mice deficient for chemokines, cytokines, as well as neurotrophic factors and their respective receptors in the MPTP model of PD, expanding our understanding of neurotrophic support and neuroinflammation. Their work explains the role of microglia-mediated neuroinflammation in MPTP-induced neurodegeneration and highlights the contribution of neurotrophic factors in slowing the progression of midbrain dopaminergic neurons [85].

Microglia priming increases susceptibility to toxins, exacerbating mDA neurodegeneration [86], with aged monkeys displaying heightened and persistent microglial reactivity after MPTP application [87]. Age-dependent microglia priming involves epigenetic modifications, as evidenced by the role of histone H3K27me3 demethylase Jumonji domain containing 3 (Jmjd3) in M2-like microglia activation [88]. Tang et al. found that the suppression of Jmjd3 inhibited M2 polarization and concurrently amplified M1 microglial inflammatory responses, resulting in widespread neuron death in vitro. Additionally, in vivo suppression of Jmjd3 in the SN markedly induced microglial overactivation and worsened dopamine neuron death in the MPTP-intoxicated mouse model of PD [88]. Furthermore, a lower level of Jmjd3 in the midbrain of aged mice, accompanied by an increased level of H3K27me3 and a higher ratio of M1 to M2 markers, suggested that aging plays a crucial role in altering microglia phenotypes [88]. Overall, these findings suggest that Jmjd3 can enhance M2 microglia polarization by modifying histone H3K27me3, thereby playing a crucial role in the switch of microglia phenotypes that may contribute to the immune pathogenesis of PD.

Aging has an influence on the severity of MPTP-induced neurodegeneration, as shown by a study utilizing senescence-accelerated mouse prone 8, a mouse strain with premature senility onset, demonstrating heightened microglial cell activation and increased neurodegeneration post-MPTP intoxication [89]. Apart from toxin-based PD models, αSyn transgenic mice are frequently employed to elucidate how αSyn aggregates contribute to neuroinflammation and neurodegeneration in PD. Non-aggregated αSyn can trigger TLR-mediated immune responses of microglia, potentially contributing to sporadic and/or familial forms of αSyn-related PD [90]. 

Scheffold et al. studied telomere shortening, a consequence of incomplete replication of chromosome ends, which is a recognized hallmark of aging [91]. However, the precise role of telomere dysfunction in neurological diseases and the aging brain remains unclear, with ongoing debate regarding its association with PD. In this study, a mouse model of PD (Thy-1 [A30P] α-synuclein transgenic mouse model) was investigated in the context of telomere shortening using the Terc knockout mouse model. It was found that α-synuclein transgenic mice with shortened telomeres (αSYN(tg/tg) G3Terc(−/−)) exhibited accelerated disease progression, resulting in significantly reduced survival. 

This expedited phenotype in mice with truncated telomeres is marked by deteriorated motor performance and enhanced formation of α-synuclein aggregate formation [91]. Quantification studies of mRNA expression and analysis of immunohistochemica revealed that in the late stages of the disease, brain stem microglia exhibited damaged responses in αSYN(tg/tg) G3Terc(−/−) microglial cell animals. These data offer initial experimental proof that telomere shortening accelerates the pathology of α-synuclein related to compromised microglial cell function in the brainstem [91]. Extracellular alpha-synuclein (αsyn) oligomers have an important role to play in PD pathogenesis. Growing evidence demonstrates that these extracellular entities activate microglia, leading to heightened neuronal impairment [92]. Despite the studies that are being carried out, little is known about the age impact on phagocytosis and microglial cell activation, particularly of extracellular αsyn oligomers. This study demonstrates that microglia isolated from adult mice, unlike those from young mice, exhibit phagocytosis deficiencies of free and exosome-associated αsyn oligomers along with increased TNFα secretion [92]. Additionally, in this work, a dysregulation of monocyte subpopulations in aging mice and humans is described. Human monocytes from elderly donors also displayed reduced phagocytic activity of extracellular αsyn. These results explain that these age-related alterations may contribute to enhanced susceptibility to pathogens or abnormally aged folded proteins in neurodegenerative diseases [92].

In summary, the involvement of aged microglia in the progressive nature of PD appears likely, especially considering the high density of microglia in the nigrostriatal system [39], further bolstering the notion of microglial involvement in PD pathogenesis. However, the molecular and functional alterations of aged microglia are only partially understood, and their role in neurodegeneration and neuroinflammation in aged individuals requires further investigation in future studies.

## 7. Therapeutic Targets for Regulating Microglial Cells in Alzheimer’s and Parkinson’s Diseases

There is a growing fascination in the search for new therapeutic approaches based on the regulation of the activity of microglia to reverse inflammation and oxidative stress. Previously described microglial modulators include various surface receptors (e.g., CD200R, PGE2 receptors) and key signaling molecules (e.g., PI3K/AKT, GSK3β, HMGB1). 

Precise and reliable quantification is essential to regulate the morphology and motility of microglial cells under physiological and disease scenarios. In a brilliant study, it was established that MotiQ v3.0 an open-source software, can be applied to in vivo, ex vivo, and in vitro data from epifluorescence, two-photon or confocal microscopy, and facilitate the automated quantification of morphology and motility parameters. These parameters are widely accepted as standards in the microglial research community [68]. The tool’s precision was validated using a benchmark data set. The analysis obtained by this technique showed retractions of ramification and tree length in plaque-distant microglia; however, plaque-associated microglia were mostly affected [93], revealing that these process extensions and retractions were slower in the AD mouse model. Notably, these data were consistent with the parameters determined using MotiQ.

Substantial evidence from experimental disease models suggests that deficiency in TLR4 or Myd88 offers neuroprotective effects by promoting the transition from M1 to M2 microglial phenotypes and reducing microgliosis [94,95]. Consequently, therapeutic strategies targeting the TLR4 pathway, such as astaxanthin and peroxiredoxin 2, have been investigated. In a model of intracerebral hemorrhage, TLR4 antagonism or genetic deletion provided neuroprotection by limiting microglial activation in perihematomal tissue. Similarly, TLR4 inhibition induces M2 polarization [96]. 

The JAK/STAT (Janus kinase) signaling cascade plays a crucial role in modulating both innate and adaptive immune responses and has been shown to influence microglial cell phenotypes in vitro and in vivo. STAT1 promotes M1 polarization in response to hypoxic stimuli by upregulating inducible nitric oxide synthase (iNOS), cyclooxygenase-2 (COX-2), and CD86 [97]. In vivo, JAK/STAT inhibition reduces M1 polarization while enhancing M2 polarization. This pathway is also involved in regulating microglial responses in pathological conditions. These findings suggest that inhibiting microglial JAK/STAT signaling may have therapeutic potential for neurodegenerative diseases [98]. 

The CX3CR1-CX3CL1 signaling axis is another crucial regulator of microglial activation and function. CX3CR1, primarily expressed by microglial cells, interacts with its neuron-secreted ligand CX3CL1 to facilitate microglia–neuron communication, acting as an essential immune checkpoint [99]. This interaction suppresses M1 pro-inflammatory responses, such as the expression of IL-1β, IL-6, TNFα, and iNOS [99]. Downregulation of either CX3CR1 or CX3CL1 is linked to increased microglial activation and neuroinflammation. This signaling pathway has been shown to be neuroprotective in various neurodegenerative models. Additionally, CX3CR1 knockout young microglia exhibit a premature aging transcriptome, altered morphology, and upregulated inflammatory pathways, indicating a protective role for CX3CR1-CX3CL1 in suppressing neurotoxic microglial activation during aging [100]. 

TREM2, a surface receptor also regulates microglial function and response to neurodegeneration, including proliferation, survival, clustering, and phagocytosis in disease states. Studies have shown that TREM2 deficiency impairs the clearance of apoptotic neurons and pathological substrates, thereby slowing disease progression [99]. Conversely, TREM2 inhibition promotes a pro-inflammatory microglial response and neurological deficits in hemorrhagic stroke models [101]. Elevated risk of developing AD is related to hypomorphic variants of TREM2. Ulland et al. investigated the role of TREM2 in the microglial response to amyloid-β pathology in AD [69,102]. Microglia in Trem2−/− mice showed reduced clustering around plaques and decreased phagocytic activity compared to wild-type mice, indicating a compromised ability to clear amyloid-β. The study combined genetic and pharmacological approaches to elucidate the function of TREM2 in disease progression. TREM2 has been considered as a critical regulator of microglial function and its loss results in altered cytokine profiles and reduced expression of genes involved in lipid metabolism and inflammatory responses in microglia. TREM2 was essential for maintaining microglial viability and preventing apoptosis in the presence of amyloid-β. Trem2−/− mice showed increased microglial cell death around plaques, indicating that TREM2 is crucial for microglial survival in AD [69,102] (Table 1). 

The crucial duty of CSF1R signaling in microglial cell homeostasis in the adult brain is unknown. In an interesting study, the effects of selective CSF1R inhibitors on microglia were tested in adult mice and the data showed that microglia in the adult brain are physiologically dependent upon CSF1R signaling. Mice depleted of microglia showed no behavioral or cognitive abnormalities, revealing that microglia are not necessary for these tasks. The microglia-depleted brain was completely repopulated through the proliferation of nestin-positive cells that then differentiated into microglia within one week of inhibitor cessation [12]. In a recent study, Spangenberg et al. explored the effects of depleting microglia and cognitive function, utilizing a selective brain-penetrant CSF1R inhibitor (PLX5622), which is essential for the survival of microglial cells [70]. They observed, in a mouse model of AD, significant reductions in amyloid plaques and improvements in cognitive performance. However, Aβ deposits in cortical blood vessels were reminiscent of cerebral amyloid angiopathy. The study revealed that the removal of microglia can mitigate amyloid-induced neurotoxicity [70] (Table 1).

Deposition of Aβ peptide drives cerebral neuroinflammation in AD by activating microglial cells, as previously mentioned. Indeed, NOD-like receptor thermal protein domain associated protein 3 (NLRP3) inflammasome activation by Aβ in microglia is crucial for interleukin-1β maturation and subsequent inflammatory processes [71]. However, the role of NRPL3 inflammasome activation in microglia and in AD in vivo remains unknown. The research of Heneka et al. demonstrated that amyloid-β activates the release of caspase-1 expression in human brains and IL-1β activation as well as enhanced Aβ clearance [71]. Furthermore, genetic or pharmacological inhibition of NLRP3 inflammasome reduced deposition of Aβ in the APP/PS1 model of AD. These results highlight NLRP3 as a potential therapeutic target for reducing neuroinflammation and slowing the progression of AD [71] (Table 1). 

Purinergic P2Y receptors play a pivotal role in modulating microglial responses to stress and injury and are significantly implicated in neurodegenerative diseases. The P2Y6 receptor (P2Y6R) is crucial for microglial phagocytosis of neurons and is mainly activated by UDP released from stressed or damaged neurons [103]. Mice lacking P2Y6R are protected from neuronal loss and cognitive deficits caused by aging and Aβ/tau pathology [104]. The P2Y12 receptor (P2Y12R) is exclusively expressed in microglial cells within the CNS and is activated by ADP derived from ATP breakdown. Its key roles include alterations in signature transcripts as microglial cells transition from a homeostatic state to a neurotoxic phenotype. This downregulation is observed in AD models and other tauopathies, with reduced expression linked to regions with dense tau aggregates and significant neurodegeneration [72]. Restoring P2Y12R expression during aging could have therapeutic benefits for neurodegenerative diseases, though further research is necessary to understand its mechanisms across different disease stages [72] (Table 1).

Ngolab et al. conducted pivotal research on the role of exosomes in patient brains diagnosed with either AD or those that contained aggregate-prone proteins. This work demonstrated that the brain-derived exosomes from patients facilitated the intercellular transfer of toxic proteins in vivo, thereby contributing to the spread of the neurodegenerative processes across different regions of the brain. α-synuclein aggregation was observed in MAP2+ and Rab5+ neurons. This insight highlights the potential of targeting exosome-mediated pathways as a therapeutic approach to mitigating the progression of AD [73] (Table 1).

A crucial issue in AD is the amyloid plaque niches. One study elucidated how glial cells collaborate to modulate the inflammatory environment and influence amyloid plaque formation and clearance. This study used two high-resolution spatial transcriptomics platforms, CosMx and Spatial-Enhanced Resolution Omics-sequencing (Stereo-seq), to characterize the transcriptomic alterations, cellular compositions, and signaling perturbations in the amyloid plaque niche in an AD mouse model [74]. The data of this work discovered that astrocytes play a supportive role in enhancing microglial phagocytic activity, thereby affecting plaque dynamics. Astrocytes secrete chemokines and cytokines the attract microglial cells to the sites of plaque deposition, inducing GABAergic signaling, decreasing glutamatergic signaling in hippocampal neurons, and inducing an imbalance in neuronal synaptic signaling [105] (Table 1).

Lee and colleagues investigated the anti-inflammatory activity of Sodium thiosulfate in our glial-mediated neuroinflammatory model and found that sodium thiosulfate increases hydrogen sulfide and glutathione expression in human microglia and astrocytes [75]. When the glial cells were treated with sodium hydrosulfide, there was a significant enhancement of neuroprotection. Although sodium hydrosulfide was somewhat more powerful than sodium thiosulfate in these in vitro assays, sodium thiosulfate has already been approved as an orally available treatment. Sodium thiosulfate may therefore be a candidate for treating neurodegenerative disorders that have a prominent neuroinflammatory component [75] (Table 1).

Peroxisome Proliferator-Activated Receptor γ (PPARγ) is a nuclear receptor in microglia that acts as a ligand-activated transcription factor involved in immune responses. The activation of PPARγ exerts neuroprotective effects such as promoting a switch from pro-inflammatory M1 to anti-inflammatory M2 microglia [106]. PPARγ activation increases M2 markers and decreases M1 markers. In AD and PD, PPARγ agonists reduce pro-inflammatory cytokine production and prevent neuronal loss. They enhance microglial amyloid clearance by upregulating scavenger receptors such as CD36, thereby reducing Aβ levels and improving cognitive functions. Clinical trials are exploring the therapeutic potential of PPARγ agonists, such as pioglitazone, in treating neurocognitive diseases due to these beneficial roles [107].

## 8. Conclusions

Microglial senescence appears to underlie the transition of microglia from being neuroprotective in the young brain to neurotoxic in the aged brain. Microglial functions have been clearly affected by aging and activation states both in vitro and in vivo. Aging-related changes in microglial activation and neuroinflammation enhance their neurotoxicity, and additionally, it seems that the distinctive nature of microglia contributes to their age-dependent functional impairment. However, it remains unclear whether aged microglia are responsible for exacerbating neurodegeneration in aged individuals or if aged neurons themselves are more susceptible to degenerative cues. The development of models for incorporating aged microglia into both in vitro and in vivo studies of neurodegenerative diseases and aging itself has a significant potential impact in pioneering the reversal of aging effects in microglia. For example, successful reversal of the iron-overloaded phenotype could not only deepen the understanding of the aging process but also facilitate the development of potent brain anti-aging therapies, offering protection against debilitating neurodegenerative diseases in older individuals.

Overall, further analysis of the effect of aging on microglia is necessary to better understand the molecular mechanisms underlying age-related changes in microglial phenotypes and functions. 

Furthermore, the onset, severity, and progression of neurodegenerative diseases such as AD and PD are influenced by aging and the aging-associated changes in microglial functions. The accumulation of senescent cells in the aging brain might create favorable proinflammatory conditions that promote the onset of AD alongside other risk factors. Subsequently, feedback loops between senescent cells and AD-related neuropathological features, such as Aβ plaques and neurofibrillary-tangle-accumulating neurons, could hasten neurodegeneration, worsening cognitive decline. Microglia exhibit significant alterations in morphology and motility in response to amyloid plaques. Different studies have demonstrated that plaque-associated microglia show reduced ramification and slower process dynamics, which likely contribute to impaired clearance of amyloid-beta and exacerbation of neuroinflammation. The next steps in the understanding of microglia are to evaluate the microglial response to the environment and to determine whether to switch their phenotypes following the early stages of an insult both the infant and aging brain, because current evidence shows that microglia with normal cellular homeostasis become aberrant and dysregulated during the aging process of the CNS, resulting in an increased susceptibility to subsequent immune challenges. In addition, genetic approaches should be applied to explore key targets contributing to the connections or activation of the microglia–synapse pathways to achieve early prevention and a potential cure for AD. These changes in microglial behavior are essential for developing targeted therapeutic strategies to modulate their activity, potentially mitigating the neurodegenerative effects of AD and improving patient outcomes.

Aging is associated with shifts in the number of non-classical monocytes and diminished phagocytosis of both free and exosome-associated α-synuclein in PD. These changes in monocytes among older adults could potentially increase susceptibility to pathogens or misfolded proteins in aging humans. Also, in different studies, a connection between microglial activation and PD has been found. α-synuclein might activate microglia through the TLR4 pathway, and microglia, in turn, caused injury to dopaminergic neurons. 

The activation of the NLRP3 inflammasome signaling complex plays a crucial role in inducing a pro-inflammatory state in microglia in response to α-synuclein. Diverse α-synuclein species produce different microglial responses via TLR ligation in PD. Additionally, the interaction between α-synuclein and NLRP3 results in amplification of the inflammatory signaling pathways, exacerbates microglia activation, and contributes to neuronal damage with degradation of DA neurons. NLRP3 can be considered as a therapeutic strategy to mitigate inflammation-mediated neurodegeneration.

Further investigation is needed to fully understand the mechanisms underlying dysregulated microglial and myeloid cells during aging and their influence on the progression of neurocognitive disorders.

Notably, research into age-related diseases beyond the central nervous system increasingly indicates causal connections between cellular senescence and disease progression. These areas of study could provide a valuable framework for delving deeper into the roles of senescence in both healthy and pathological brain aging.

Targeting the regulation of microglial activation emerges as a promising therapeutic strategy, as inhibiting them can ameliorate some neurodegenerative changes. However, further research is necessary to comprehend the regulation of beneficial or deleterious microglial activation in disease progression. Rather than a broad inhibition of microglia, the therapeutic aim should focus on promoting their protective functions while mitigating their harmful effects.

## Figures and Tables

**Figure 1 biomedicines-12-01737-f001:**
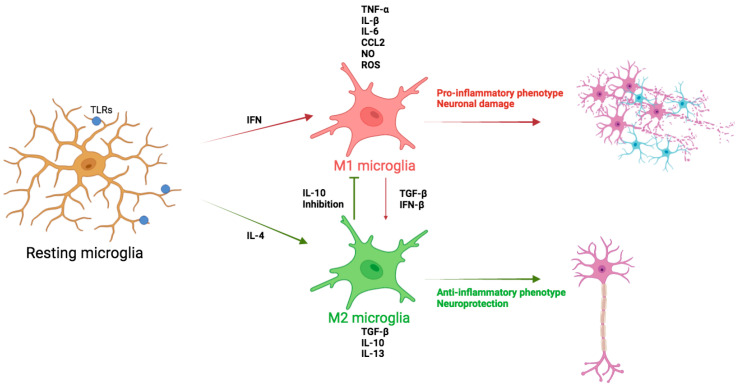
Schematic overview of the intricate relationship between microglia and neuroinflammation. Microglia become activated (M1) due to aging, oxidative stress, and different factors, which can lead to neuroinflammation and neurodegeneration. Activated microglia produce excessive ROS, ILs, TNF-α, and others, triggering neuroinflammation to promote neuronal damage and cell death. The M2 microglial phenotype contributes to the processes of phagocytosis and neuronal survival.

**Figure 2 biomedicines-12-01737-f002:**
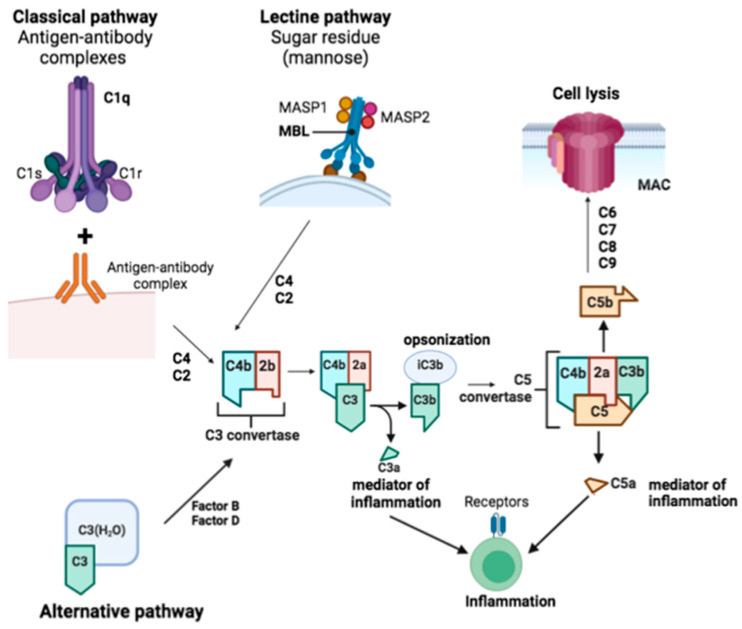
Diagram of the complement cascade. The complement can be activated by three separate pathways: the classical, lectin, and alternative pathways. All of these routes lead to the cleavage of C3 and subsequently C5, leading to the opsonization of tissues, the production of C3a and C5a, and the assembly of the cytolytic membrane complex.

**Table 1 biomedicines-12-01737-t001:** Recent advancements in the therapeutic potential of microglia on Parkinson’s and Alzheimer’s diseases.

Advancement	Description	Disease	Source/Study
Microglia Activation Modulation	-Strategies to modulate microglial activation to reduce neuroinflammation and promote a neuroprotective phenotype	Alzheimer’s	Hansen et al., 2022 [68] https://doi.org/10.1091/mbc.E21-11-0585
Targeting TREM2	-Therapeutic targeting of TREM2 to enhance microglial response and clearance of amyloid-beta plaques	Alzheimer’s	Ulland et al., 2017 [69] http://doi.org/10.1016/j.cell.2017.07.023
Colony-Stimulating Factor1 Receptor (CSF1R) Inhibitors	-Use of CSF1R inhibitors to deplete microglia and reset the microglial population	Alzheimer’s	Elmore et al., 2014 [12] http://doi.org/10.1016/j.neuron.2014.02.040
	-Sustained microglial depletion with CSF1R inhibitor impairs parenchymal plaque development in an Alzheimer’s disease model.	Parkinson’s	Spangenberg et al. 2019 [70] https://doi.org/10.1038/s41467-019-11674-z
NLRP3 Inflammasome Inhibition	-Inhibition of NLRP3 inflammasome to reduce neuroinflammation and amyloid-beta pathology	Alzheimer’s	Heneka et al., 2013 [71] https://doi.org/10.1038/nature11729
P2Y12 Receptor Antagonists	-Blocking P2Y12 receptors to modulate microglial activity and reduce neuroinflammation	Parkinson’s	Moore et al., 2015 [72] https://doi.org/10.1212/NXI.0000000000000080
Microglia-Derived Extracellular Vesicles	-Utilizing microglia-derived extracellular vesicles for targeted delivery of therapeutic agents	Parkinson’s	Ngolab et al., 2021 [73] https://doi.org/10.1186/s40478-017-0445-5
Gene Therapy	-Gene Therapy approaches to enhance microglial clearance of pathological proteins	Alzheimer’s	Sierksma et al., 2020 [74] http://doi.org/10.1126/science.abb8575
Anti-inflammatory Drugs	-Development of anti-inflammatory drugs targeting microglial pathways to reduce neuroinflammation	Alzheimer’s Parkinson’s	Lee et al., 2016 [75] https://doi.org/10.1186/s12974-016-0488-8

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
