# Peer review of "Aged-Related Changes in Microglia and Neurodegenerative Diseases: Exploring the Connection"

_biomedicines, 2024, doi:10.3390/biomedicines12081737_

Round 1

Reviewer 1 Report

Comments and Suggestions for Authors

In this review, the author bring together the current knowledge on functional and phenotypic properties of aging microglia in different conditions and pathological situations. They used proper literature and concentrated on contribution of aged microglia to the development and progression of neurodegenerative diseases such as Alz- heimer’s Disease (AD) and Parkinson’s disease (PD). However, there are still more room to be revised extensively:

1. Is there any functional consequences of the

presence of different shaped microglia in the

different CNS regions? This needs to include

2. In the chapter discussing the relationship of age

and microglia shape and function, some kind of

chronological order would be nice: embryonic,

childhood, normal adult function, normal elderly,

neurodegeneration... their provided information is signiicantly missed

3. What can be the signaling pathway of the chronic

stress effect on neurons?

The difference between males and females could be

more complicated than only hormones. What other

factors should be mentioned in addition to the

hormonal levels?

4. Also, the different “subtypes” of microglia morphology/aged

need to be described in details, and also functional

consequences. In the chapters

describing the three prominent neurodegenerative

diseases the different shapes of microglia cells are

listed, but it would be nice, if we can see those in the

tables

5. What is the originality and necessity of this review

compared to PMID: 37735487; PMID: 35805174;

PMID: 35219055; PMID: 36593381;

6. It would be great to link their perspective to discuss how microglia phenotypes and

morphological diversity/aged in Neuroinflammation

on therapeutic strategies 

7. The author aimed to review MG in neurodegenerative disesases (NDs), but how about the other diseases including multiple system atrophy (MSA), amyotrophic lateral sclerosis (ALS), frontotemporal dementia (FTD), progressive supranuclear palsy (PSP), corticobasal degeneration (CBD), dementia with Lewy bodies (DLB)? All needs to be involved

8. Include one additional table regarding recent advancements on the therapeutic potential of MG on NDs in this table (pre/clinical studies).

9. Finally, an additional chapter like "Microglia as therapeutic strategies.

target" will make their work more interesting, but it still needs to be

expanded and discussed thoroughly since they wanted to apply these mechanisms is crucial for developing better preventive and therapeutic strategies

Author Response

In this review, the author bring together the current knowledge on functional and phenotypic properties of aging microglia in different conditions and pathological situations. They used proper literature and concentrated on contribution of aged microglia to the development and progression of neurodegenerative diseases such as Alzheimer’s Disease (AD) and Parkinson’s disease (PD). However, there are still more room to be revised extensively:
-Thank you for your comments and suggestions that allowed me to greatly improve the quality of the manuscript. I agree with all your clever comments. 

1. Is there any functional consequences of the presence of different shaped microglia in the different CNS regions? This needs to include.
- Thank you for your very careful review of our paper. I have added this information that was requested by the reviewer.
The text added in red in the manuscript (page 3):
“In an interesting study, an immunophenotypic variation in brain microglial cells had been observed and suggested, noting that cortical and striatal microglia are similar, while hippocampal microglia display an profile between pro- and anti-inflammatory states [36]. Grabert et al., described how the microglial cells in the hippocampus decrease the expression of genes related to environment sensing, provoking more vulnerable to aging and disease related protein deposition [36]. It has also been observed that in healthy conditions, microglial cells in the hippocampus present a higher “immune-vigilant” phenotype. This can be related to a higher chronic inflammatory microglial response in AD pathology to plaque formation. Another particular features of the hippocampus that might influence microglial activity, are microglial cells density and the gray matter content [37]. The fractalkine receptor, CXCR3 has been shown to be involved in neuron-microglia communication, so that, hippocampal lower values of CXCR3 might provoke less microglial cells control [37]. These changes may influence microglial cells activity patterns and predispose this activity towards hippocampal-related neurocognitive diseases, where neuroinflammation and microglial cells activation and neuroinflammation play crucial roles [37]
   Temporal lobes including motor areas and the prefrontal cortex are extensively connected, playing important roles in top-down behavioral control and language processing exhibing the highest degree of age-related atrophy, in the case of prefrontal cortex, with a significant decline in prefrontal gray matter volume compared to other areas such as superior parietal cortices and the inferior temporal. In the healthy cortex [37]. Microglia/macrophage-specific inflammatory receptors CD11b expression on microglia is consistently lower than in the spinal cord, and the microglia protein, CD40,  expression is lower in the cortex compared to the cerebellum. Conversely, CXCR3 expression is higher in the cortex than in the cerebellum [37].
   Also, microglia constitute 12% of the cellular population in the substantia nigra, a particularly dense population [37]. CXCR3 is more abundant in the striatum than in other structures like the cerebellum [37]. Another important receptor in microglia-neuron interactions is the membrane glycoprotein CD200, which is downregulated with age in the substantia nigra [37]. A high density of microglial cells, reduced CD200 receptor expression, or a decreased dopaminergic population can diminish the regulatory influence of neurons on microglial cells. Such an environment may lead to a reduction in the production of trophic factors or an increase in inflammatory molecules, which can be detrimental over time.”
Also, additional references have been added and appear in red (as the new text) in the manuscript:
36. Grabert K., Michoel T., Karavolos M. H., Clohisey S., Baillie J. K., Stevens M. P., et al. (2016). Microglial brain region-dependent diversity and selective regional sensitivities to aging. Nat. Neurosci. 19 504–516. http://doi.org/10.1038/nn.4222
37. Bachiller S, Jiménez-Ferrer I, Paulus A, Yang Y, Swanberg M, Deierborg T, Boza-Serrano A. Microglia in Neurological Diseases: A Road Map to Brain-Disease Dependent-Inflammatory Response. Front Cell Neurosci. 2018 Dec 18;12:488. http://doi./10.3389/fncel.2018.00488

2. In the chapter discussing the relationship of age and microglia shape and function, some kind of chronological order would be nice: embryonic, childhood, normal adult function, normal elderly, neurodegeneration... their provided information is signiicantly missed.
- As you proposed, I have added the sentences that show the relationship of age and microglia shape and function.
The text added in red in the manuscript (page 2):
“ The use of colony-stimulating factor-1 receptor (CSF1R) inhibitors has provided significant insights into microglial dynamics in the adult CNS. In a interesant work, Elmore et al. in 2014 [12] observed that administering CSF1R inhibitors almost completely eliminated microglial cells in the adult CNS. Following the withdrawal of these inhibitors, microglia rapidly repopulated the CNS. This repopulation was associated with an increase in nestin-positive cells throughout the CNS, suggesting these cells represent microglial progenitors [12]. Initially, it was proposed that the repopulating microglia originated from a neuroectodermal lineage. This hypothesis was based on the observation that microglial cells are myeloid lineage cells and nestin-positive progenitor cells originate from the neuroectodermal lineage [13] suggested that repopulated microglia are derived from myeloid progenitor cells. However, recent findings challenge these theories. In a recent study, an adult mouse model have been used and it has been demonstrated that repopulated microglia do not originate from blood cells, nestin-positive cells, astrocytes, oligodendrocyte precursor cells, or neurons. Instead, after the selective elimination of more than 99% of microglia, the remaining microglia (< 1%) proliferated to repopulate the entire brain. Thus, the surviving microglia are the sole source of the new microglia, which rapidly repopulate the brain [13]. In aged mice, the cortex contains fewer microglia compared to young adult mice, and these cells are smaller, less symmetrical, more elongated, and have fewer branches [13]. Aging affects microglia through telomere shortening, DNA damage, and oxidative stress. Telomeres, which are the protective ends of chromosomes, shorten with age, correlating with a decline in microglial self-renewal [14]. Aged microglia express higher levels of tumor necrosis factor-α (TNF-α) and    interleukin IL-6, and their dystrophic morphology hinders spatial learning [13]. Senescent microglia show increased levels of pro-inflammatory cytokines, reduced levels of chemokines, and decreased phagocytosis of amyloid beta (Aβ) fibrils [13]. They also exhibit a 'primed' phenotype, characterized by an exaggerated and uncontrolled inflammatory response to immune stimuli [13].”
The references added in red in the text:
12. Elmore, M. R., Najafi, A. R., Koike, M. A., Dagher, N. N., Spangenberg, E. E., Rice, R. A., Kitazawa, M., Matusow, B., Nguyen, H., West, B. L., & Green, K. N. (2014). Colony-stimulating factor 1 receptor signaling is necessary for microglia viability, unmasking a microglia progenitor cell in the adult brain. Neuron, 82(2), 380–397. https://www.ncbi.nlm.nih.gov/pmc/articles/PMC4161285/
13. Xu Y, Jin MZ, Yang ZY, Jin WL. Microglia in neurodegenerative diseases. Neural Regen Res. 2021 Feb;16(2):270-280. http://doi.org/10.4103/1673-5374.290881
14. Wolf, S. A., Boddeke, H. W., & Kettenmann, H. (2017). Microglia in Physiology and Disease. Annual review of physiology, 79, 619–643. https://doi.org/10.1146/annurev-physiol-022516-034406

3. What can be the signaling pathway of the chronic stress effect on neurons? The difference between males and females could be more complicated than only hormones. What other factors should be mentioned in addition to the hormonal levels?
- In accordance with Reviewer’s question, I have redacted a summary about these topics. Text and referenced added (appear in red in the final version of the paper) (page 9 and 8, respectively):
“mTORC1 signaling has been also identified as a promising target for the treatment of major depressive disorder. Luo et al. explored whether the mTORC1 signaling pathway is involved in synapse loss in the hippocampus caused by chronic stress. After successfully establishing the chronic restraint stress model, significant changes in the mRNA levels of certain immediate early genes were observed, indicating neuronal activation and changes in protein synthesis [69]. There was a notable downregulation of glutamate receptors and postsynaptic density protein 95 at both the mRNA and protein levels. Synaptic fractionation assays indicated that chronic stress resulted in synapse loss in both ventral and dorsal hippocampus. These effects were related to the mTORC1 signaling pathway, as protein synthesis and phosphorylation of downstream signaling targets were reduced following chronic stress. Moreover, intracerebroventricular infusion of rapamycin induced depression-like behaviors and inhibited the antidepressant effects of fluoxetine [69]. This study demonstrates that the mTORC1 signaling pathway plays a crucial role in mediating synapse loss provoked by chronic stress and contributes to the behavioral effects of antidepressant treatment”
69. Luo YF, Ye XX, Fang YZ, Li MD, Xia ZX, Liu JM, Lin XS, Huang Z, Zhu XQ, Huang JJ, Tan DL, Zhang YF, Liu HP, Zhou J, Shen ZC. mTORC1 Signaling Pathway Mediates Chronic Stress-Induced Synapse Loss in the Hippocampus. Front Pharmacol. 2021 Dec 20;12:801234. http://doi./10.3389/fphar.2021.801234

“Distinct phenotypes have been identified based on sex and brain regions. An age-related increase in senescence-associated genes and higher expression of cyclin-dependent kinase inhibitors, such as Cdkn2a/p16INK4, that were specifically noted in the hippocampus of older female mice [59]. The highest occurrence of p16-positive brain cells was found in monocytes and microglial cells populations. Transcriptomic profiling of these two myeloid clusters revealed an upregulation of chemoattractant SASP factors such as secreted phosphoprotein 1 (Spp1), MHC genes, lysosomal stress markers (Lgals3 and Lgals3bp) and senescence-related genes (Cdkn2a, Cdkn1a, and B cell Lymphoma (Blc2)). Although sex influenced diverse age-related signatures in the aging hippocampus, different works indicated that systemic clearance of p16-positive cells improved cognitive functions in neurodegenerative conditions [5]. Consistent with previousstudies, spatially resolved transcriptomic analysis in brain sections of aged animals confirmed the age-related increase of Cdkn2a expression. Furthermore, the microdomains containing senescent cells were heterogeneously distributed in the cortical gray matter, hippocampus and white matterand were mainly represented by clusters of cells expressing endothelial cells, markers of microglial cells, oligodendrocytes progenitor cells and oligodendrocytes [60].”
59. Gorgoulis, V., Adams, P. D., Alimonti, A., Bennett, D. C., Bischof, O., Bishop, C., Campisi, J., Collado, M., Evangelou, K., Ferbeyre, G., Gil, J., Hara, E., Krizhanovsky, V., Jurk, D., Maier, A. B., Narita, M., Niedernhofer, L., Passos, J. F., Robbins, P. D., Schmitt, C. A., … Demaria, M. (2019). Cellular Senescence: Defining a Path Forward. Cell, 179(4), 813–827. https://doi.org/10.1016/j.cell.2019.10.005
60. Bussian, T. J., Aziz, A., Meyer, C. F., Swenson, B. L., van Deursen, J. M., & Baker, D. J. (2018). Clearance of senescent glial cells prevents tau-dependent pathology and cognitive decline. Nature, 562(7728), 578–582. https://doi.org/10.1038/s41586-018-0543-y

4. Also, the different “subtypes” of microglia morphology/aged need to be described in details, and also functional consequences. In the chapters describing the three prominent neurodegenerative diseases the different shapes of microglia cells are listed, but it would be nice, if we can see those in the tables. 
- I have followed the Reviewer’s recommendation and have added the information.
“In the presence of pathology, microglial cells changes both their shape and function, undergoing a transformation from ramified to hypertrophic and ameboid phenotypes-a progression that continues to guide research today [31]. The build-up of senescent microglia, along with their impaired immune functions and interactions with other brain cells, could play a crucial role in the development of neurodegenerative diseases [31]. The unique characteristics of yolk sac-derived microglial cells, which have a limited capacity for repopulation, suggest an intrinsic link to their senescence, potentially contributing significantly to age-related neurodegenerative diseases [31]. Regarding the satellite microglia, this subgroup is now identified by its distinctive association with neuronal cell bodies [32]. The portion of the axon and the satellite microglial cells overlaps and action potentials are initiated [32]. It has been observed both during development and adulthood in mice, with a preferential association with excitatory neurons [32] and additionally, this subpopulation has been noted in the cerebral cortex of adult rats and adult non-human primates using non-invasive two-photon in vivo imaging [33], indicating conservation across species. These cells show soma migration, interacting with both Purkinje neuronal cell bodies and proximal dendrites, are less ramified than their cortical counterparts, and exhibit reduced surveillance. This finding suggests that satellite microglia are heterogeneous in their dynamics under steady-state conditions [33]. Beyond the distinct morphological properties of microglia revealed by light or fluorescence microscopy, high-resolution electron microscopy has uncovered the presence of microglia filled with cellular debris, similar to the fat granule or gitter cells initially described by Río-Hortega (1920), during aging, age-related sensory loss, and Werner syndrome in mice [34]. Recently, electron microscopy also identified a unique microglial population, the "dark" microglia, in the adult and aged mouse hippocampus (CA1 region and dentate gyrus), cerebral cortex, amygdala, and hypothalamus. These cells coexist with the typical microglia but display several ultrastructural features, including their size, morphology, long stretches of endoplasmic reticulum, interactions with neurons and synapses, and association with the extracellular space. Unlike typical microglia, they exhibit markers of oxidative stress, such as a condensed, electron-dense cytoplasm and nucleoplasm, giving them a dark appearance in EM, along with Golgi apparatus/endoplasmic reticulum dilation, mitochondrial alterations, and a partial to complete loss of heterochromatin pattern [34]. In neurons, changes to the heterochromatin pattern are associated with cellular stress, aging, and brain disorders such as schizophrenia and Alzheimer's disease [35].”
The reference added in red in the text:
31. Rim C, You MJ, Nahm M, Kwon MS. Emerging role of senescent microglia in brain aging-related neurodegenerative diseases. Transl Neurodegener. 2024 Feb 20;13(1):10. http://doi.org/10.1186/s40035-024-00402-3
32. Wogram, E., Wendt, S., Matyash, M., Pivneva, T., Draguhn, A., & Kettenmann, H. (2016). Satellite microglia show spontaneous electrical activity that is uncorrelated with activity of the attached neuron. European Journal of Neuroscience, 43(11), 1523-1534. https://doi.org/10.1111/ejn.13256
33. Stowell, R. D., Wong, E. L., Batchelor, H. N., Mendes, M. S., Lamantia, C. E., Whitelaw, B. S., & Majewska, A. K. (2018). Cerebellar microglia are dynamically unique and survey Purkinje neurons in vivo. Developmental neurobiology, 78(6), 627–644. https://doi.org/10.1002/dneu.22572
34. Hui, B., Zhang, L., Zhou, Q., & Hui, L. (2018). Pristimerin Inhibits LPS-Triggered Neurotoxicity in BV-2 Microglia Cells Through Modulating IRAK1/TRAF6/TAK1-Mediated NF-κB and AP-1 Signaling Pathways In Vitro. Neurotoxicity research, 33(2), 268–283. https://doi.org/10.1007/s12640-017-9837-3
35. Medrano-Fernández, A., & Barco, A. (2016). Nuclear organization and 3D chromatin architecture in cognition and neuropsychiatric disorders. Molecular brain, 9(1), 83. https://doi.org/10.1186/s13041-016-0263-x

5. What is the originality and necessity of this review compared to PMID: 37735487; PMID: 35805174; PMID: 35219055; PMID: 36593381
- Thank you for your very accurate, meticulous and careful review of my paper. I am convinced the document has been considerably improved. In this case, I considered that all these excellent papers you mention are very much focused on the clinical role of microglia in different diseases. My modest paper is perhaps more focused on the mechanisms and role of microglia in inflammation, oxidative stress…
In Wendimu's great paper (referenced in my review) they focuse on phenotypes of microglia in neurodegenerative diseases and in the case of Mahmood et al., they focus on remyelination and not so much on the specific role and mechanisms of aged microglia in different diseases. Also Gao et al. show in their paper the role of microglia in neurodegenerative diseases: mechanism and potential therapeutic targets but not specifically in aged microglial cells.

6. It would be great to link their perspective to discuss how microglia phenotypes and morphological diversity/aged in Neuroinflammation on therapeutic strategies 
- According with the smart Reviewer’s comment, I have revised and linked these topics in page 12 (new section 7) and have added the new reference (text in red): 
“There is significant interest in identifying therapeutic targets that can modulate microglial activity to promote protective anti-inflammatory and phagocytic phenotypes while suppressing pro-inflammatory and neurotoxic responses. Previously described microglial modulators include various surface receptors (e.g., CD200R, PGE2 receptors) and key signaling molecules (e.g., PI3K/AKT, GSK3β, HMGB1). 
To control microglial morphology and motility under physiological and disease conditions, it is necessary to quantify microglial motility and morphology precisely and reliably. Hansen et al. have developed MotiQ, an open-source, freely accessible software for automated quantification of microglial morphology and motility. MotiQ allows quantification of morphology and motility parameters that have become the gold standard in the microglia field. It has been shown that MotiQ can be applied to in vivo, ex vivo, and in vitro data from epifluorescence, two-photon or confocal microscopy [86]. The accuracy and applicability of this tool has been assessed on a benchmark data set. MotiQ 3D analysis shown a significant reduction of ramification and tree length in plaque-associated and plaque-distant microglia, while plaque-associated microglia were mostly affected, in line with previous reports in another AD mouse model [87]. This report showed that process extensions and retractions were slower in the AD mouse model, especially at the plaque. Importantly, the results from manual tracking correlated with the motility parameters determined with MotiQ analysis.”
86. Hansen, J. N., Brückner, M., Pietrowski, M. J., Jikeli, J. F., Plescher, M., Beckert, H., Schnaars, M., Fülle, L., Reitmeier, K., Langmann, T., Förster, I., Boche, D., Petzold, G. C., & Halle, A. (2022). MotiQ: an open-source toolbox to quantify the cell motility and morphology of microglia. Molecular biology of the cell, 33(11), ar99. 
87. Plescher, M., Seifert, G., Hansen, J. N., Bedner, P., Steinhäuser, C., & Halle, A. (2018). Plaque-dependent morphological and electrophysiological heterogeneity of microglia in an Alzheimer's disease mouse model. Glia, 66(7), 1464–1480. https://doi.org/10.1002/glia.23318

7. The author aimed to review MG in neurodegenerative disesases (NDs), but how about the other diseases including multiple system atrophy (MSA), amyotrophic lateral sclerosis (ALS), frontotemporal dementia (FTD), progressive supranuclear palsy (PSP), corticobasal degeneration (CBD), dementia with Lewy bodies (DLB)? All needs to be involved
- In accordance with intelligent Reviewer’s comment, I reply that my initial idea was to focus on the role of microglia in Parkinson's disease (on which I have personally worked and published papers years ago) and Alzheimer's disease, giving examples and specific studies carried out on them. I think that including the specificities of each of the diseases mentioned here would be the subject of another review. I will keep this in mind to continue with my studies of microglia extended to other diseases and for my next publications.

8. Include one additional table regarding recent advancements on the therapeutic potential of MG on NDs in this table (pre/clinical studies).
- According with the review’s comment I have added a table (Table 1) that expresses the different advances on the therapeutic potential of microglia in diseases such as AD and PD.

9. Finally, an additional chapter like "Microglia as therapeutic strategies. target" will make their work more interesting, but it still needs to be expanded and discussed thoroughly since they wanted to apply these mechanisms is crucial for developing better preventive and therapeutic strategies
- Thanks a lot for your very accurate review of my manuscript. I have added the additional table (8) (page 15) and the additional chapter (9).
The added text in red in page 13. The additional references have been added in red in the manuscript.
“Substantial evidence from experimental disease models suggests that deficiency in TLR4 or Myd88 offers neuroprotective effects by promoting the transition from M1 to M2 microglial phenotypes and reducing microgliosis [88, 89]. Consequently, therapeutic strategies targeting the TLR4 pathway, such as astaxanthin and peroxiredoxin 2, have been investigated. In an model of intracerebral hemorrhage, TLR4 antagonism or genetic deletion provided neuroprotection by limiting microglial activation in perihematomal tissue. Similarly, TLR4 inhibition induces M2 polarization [90]. 
JAK/STAT (Janus kinase) signaling cascade plays a crucial role in modulating both innate and adaptive immune responses and has been shown to influence microglial cell phenotypes in vitro and in vivo. STAT1 promotes M1 polarization in response to hypoxic stimuli by upregulating inducible nitric oxide synthase (iNOS), cyclooxygenase-2 (COX-2) and CD86 [91]. In vivo, JAK/STAT inhibition reduces M1 polarization while enhancing M2 polarization. This pathway is also involved in regulating microglial responses in pathological conditions. These findings suggest that inhibiting microglial JAK/STAT signaling may have therapeutic potential for neurodegenerative diseases [92]. 
CX3CR1-CX3CL1 Signaling The CX3CR1-CX3CL1 signaling axis is another crucial regulator of microglial activation and function. CX3CR1, primarily expressed by microglial cells, interacts with its neuron-secreted ligand CX3CL1 to facilitate microglia-neuron communication, acting as an essential immune checkpoint [93]. This interaction suppresses M1 pro-inflammatory responses, such as the expression of IL-1β, IL-6, TNFα, and iNOS [93]. Downregulation of either CX3CR1 or CX3CL1 is linked to increased microglial activation and neuroinflammation. This signaling pathway has been shown to be neuroprotective in various neurodegenerative models. Additionally, CX3CR1 knockout young microglia exhibit a premature aging transcriptome, altered morphology, and upregulated inflammatory pathways, indicating a protective role for CX3CR1-CX3CL1 in suppressing neurotoxic microglial activation during aging [94]. 
TREM2, a surface receptor also regulates microglial function and response to neurodegeneration, including proliferation, survival, clustering, and phagocytosis in disease states. Studies have shown that TREM2 deficiency impairs the clearance of apoptotic neurons and pathological substrates, thereby slowing disease progression [93]. Conversely, TREM2 inhibition promotes a pro-inflammatory microglial response and neurological deficits in hemorrhagic stroke models [95]. Elevated risk of developing AD is related to hypomorphic variants of TREM2. In a recent study, the authors have find that microglia in AD patients carrying TREM2 risk variants and TREM2-deficient mice with AD-like pathology have abundant autophagic vesicles, as do TREM2-deficient macrophages under growth-factor limitation or endoplasmic reticulum stress [96]. Combined metabolomics and RNA sequencing linked this anomalous autophagy to defective mammalian target of mTOR signaling, which affects ATP levels and biosynthetic pathways. Dectin-1, a receptor that elicits TREM2-like intracellular signals, and cyclocreatine, a creatine analog that can supply ATP have been studied. Dietary cyclocreatine tempered autophagy, restored microglial clustering around plaques, and decreased plaque-adjacent neuronal dystrophy in TREM2-deficient mice with amyloid-β pathology. Thus, TREM2 enables microglial responses during AD by sustaining cellular energetic and biosynthetic metabolism [97]. 
The crucial duty of CSF1R signaling in microglial cells homeostasis in the adult brain is unknown. In a interesant study, the effects of selective CSF1R inhibitors on microglia have been tested in adult mice and the data have shown that microglia in the adult brain is physiologically dependent upon CSF1R signaling. Mice depleted of microglia showed no behavioral or cognitive abnormalities, revealing that microglia are not necessary for these tasks. The microglia-depleted brain completely repopulated through proliferation of nestin positive cells that then differentiate into microglia within one week of inhibitor cessation [12]. Microglial cells are dependent on CSF1R signaling for their survival. Spangenberg et al. synthesized a highly selective brain-penetrant CSF1R inhibitor (PLX5622) allowing for extended and specific microglial elimination, preceding and during pathology development [98]. They demonstrated in mouse model of AD, plaques fail to form in the parenchymal space following microglial depletion, except in areas containing surviving microglia. However, Aβ deposits in cortical blood vessels reminiscent of cerebral amyloid angiopathy. Altered gene expression in the mouse hippocampus is also reversed by the absence of microglial cells. Transcriptional analyses of the residual plaque-forming microglia showed that they exhibit a disease-associated microglia profile, and therefore, PLX5622, allowed for sustained microglial depletion and identified roles of microglial cells in initiating plaque pathogenesis [98].
Deposition of Aβ peptide drives cerebral neuroinflammation in AD by activating microglial cells as previously mentioned. Indeed, NLRP3 inflammasome activation by Aβ in microglia is crucial for interleukin-1β maturation and subsequent inflammatory process [99]. However, it remains unknown whether NLRP3 activation contributes to Alzheimer's disease in vivo. Heneka et al have tried to demonstrate the enhanced active caspase-1 expression in human brains with AD, suggesting a crucial role for the inflammasome in this disease. Nlrp3(-/-) or Casp1(-/-) mice carrying mutations demonstrated reduced brain caspase-1 and IL-1β activation as well as enhanced Aβ clearance [99]. Furthermore, NLRP3 inflammasome deficiency skewed microglia to an M2 phenotype and resulted in the decreased deposition of Aβ in the APP/PS1 model of AD. These results show an important role for the NLRP3/caspase-1 axis in the pathogenesis of AD [99]. 
Purinergic P2Y receptors play a pivotal role in modulating microglial responses to stress and injury and are significantly implicated in neurodegenerative diseases. P2Y6 Receptor (P2Y6R) P2Y6R is crucial for microglial phagocytosis of neurons and is mainly activated by UDP released from stressed or damaged neurons [100]. Mice lacking P2Y6R are protected from neuronal loss and cognitive deficits caused by aging and Aβ/tau pathology [101]. P2Y12 Receptor (P2Y12R) P2Y12R is exclusively expressed in microglial cells within the CNS and is activated by ADP derived from ATP breakdown. Its key roles include alterations in signature transcripts as microglial cells transition from a homeostatic state to a neurotoxic phenotype. This downregulation is observed in AD models and other tauopathies, with reduced expression linked to regions with dense tau aggregates and significant neurodegeneration [102]. Restoring P2Y12R expression during aging could have therapeutic benefits for neurodegenerative diseases, though further research is necessary to understand its mechanisms across different disease stages [102].
Ngolab et al, in this work, studied if the exosomes extracted from brains diagnosed with either AD contained aggregate-prone proteins. Furthermore, injection of brain-derived exosomes from patients into the brains of wild type mice induced α-synuclein aggregation. α-synclein aggregation was observed in MAP2+, Rab5+ neurons. Using a neuronal cell line, the authors also identified intracellular α-synuclein aggregation mediated by exosomes is dependent on recipient cell endocytosis. Then, these data indicate that exosomes from patients are sufficient for seeding and propagating α-synuclein aggregation in vivo [103].
The amyloid plaque niche is a pivotal hallmark of AD. In a interesant work they were employed, two high-resolution spatial transcriptomics platforms, CosMx and Spatial Enhanced Resolution Omics-sequencing (Stereo-seq), to characterize the transcriptomic alterations, cellular compositions, and signaling perturbations in the amyloid plaque niche in an AD mouse model [104]. The data about the study of transcriptomic alterations of glial cells in the vicinity of plaques indicates that the microglial response to plaques is consistent across different brain regions, while the astrocytic response is more heterogeneous. Meanwhile, as the microglial density of plaque niches increases, astrocytes acquire a more neurotoxic phenotype and play a key role in inducing GABAergic signaling and decreasing glutamatergic signaling in hippocampal neurons [104]. in this work, it is showed that the accumulation of microglia around hippocampal plaques disrupts astrocytic signaling, in turn inducing an imbalance in neuronal synaptic signaling [104].
The group of McGeer investigated the anti-inflammatory activity of Sodium thiosulfate in our glial-mediated neuroinflammatory model and found that Sodium thiosulfate increases hydrogen sulfide and glutathione expression in human microglia and astrocytes [105]. When the glial cells were treated with sodium hydrosulfide, there was a significant enhancement of neuroprotection. Although sodium hydrosulfide was somewhat more powerful than Sodium thiosulfate in these in vitro assays, Sodium thiosulfate has already been approved as an orally available treatment. Sodium thiosulfate may therefore be a candidate for treating neurodegenerative disorders that have a prominent neuroinflammatory component [105].
Peroxisome Proliferator-Activated Receptor γ (PPARγ) is a nuclear receptor in microglia that acts as a ligand-activated transcription factor involved in immune responses. Activation of PPARγ exerts neuroprotective effects such as promoting a switch from pro-inflammatory M1 to anti-inflammatory M2 microglia [106]. PPARγ activation increases M2 markers and decreases M1 markers. In AD and PD, PPARγ agonists reduce pro-inflammatory cytokine production and prevent neuronal loss. They enhance microglial amyloid clearance by upregulating scavenger receptors like CD36, thereby reducing Aβ levels and improving cognitive functions. Clinical trials are exploring the therapeutic potential of PPARγ agonists, such as pioglitazone, in treating neurocognitive diseases due to these beneficial roles [107].

88. Yao, X., Liu, S., Ding, W., Yue, P., Jiang, Q., Zhao, M., Hu, F., & Zhang, H. (2017). TLR4 signal ablation attenuated neurological deficits by regulating microglial M1/M2 phenotype after traumatic brain injury in mice. Journal of neuroimmunology, 310, 38–45. https://doi.org/10.1016/j.jneuroim.2017.06.006
89. Liu, J. T., Wu, S. X., Zhang, H., & Kuang, F. (2018). Inhibition of MyD88 Signaling Skews Microglia/Macrophage Polarization and Attenuates Neuronal Apoptosis in the Hippocampus After Status Epilepticus in Mice. Neurotherapeutics : the journal of the American Society for Experimental NeuroTherapeutics, 15(4), 1093–1111. https://doi.org/10.1007/s13311-018-0653-0
90. Yang, Z., Liu, B., Zhong, L., Shen, H., Lin, C., Lin, L., Zhang, N., & Yuan, B. (2015). Toll-like receptor-4-mediated autophagy contributes to microglial activation and inflammatory injury in mouse models of intracerebral haemorrhage. Neuropathology and applied neurobiology, 41(4), e95–e106. https://doi.org/10.1111/nan.12177
91. Jain, M., Singh, M. K., Shyam, H., Mishra, A., Kumar, S., Kumar, A., & Kushwaha, J. (2021). Role of JAK/STAT in the Neuroinflammation and its Association with Neurological Disorders. Annals of neurosciences, 28(3-4), 191–200. https://doi.org/10.1177/09727531211070532
92. Butturini, E., Boriero, D., Carcereri de Prati, A., & Mariotto, S. (2019). STAT1 drives M1 microglia activation and neuroinflammation under hypoxia. Archives of biochemistry and biophysics, 669, 22–30. https://doi.org/10.1016/j.abb.2019.05.011
93. Pawelec, P., Ziemka-Nalecz, M., Sypecka, J., & Zalewska, T. (2020). The Impact of the CX3CL1/CX3CR1 Axis in Neurological Disorders. Cells, 9(10), 2277. https://doi.org/10.3390/cells9102277
94. Bolós, M., Llorens-Martín, M., Perea, J. R., Jurado-Arjona, J., Rábano, A., Hernández, F., & Avila, J. (2017). Absence of CX3CR1 impairs the internalization of Tau by microglia. Molecular neurodegeneration, 12(1), 59. https://doi.org/10.1186/s13024-017-0200-1
95. Zhao, Y., Wu, X., Li, X., Jiang, L. L., Gui, X., Liu, Y., Sun, Y., Zhu, B., Piña-Crespo, J. C., Zhang, M., Zhang, N., Chen, X., Bu, G., An, Z., Huang, T. Y., & Xu, H. (2018). TREM2 Is a Receptor for β-Amyloid that Mediates Microglial Function. Neuron, 97(5), 1023–1031.e7. https://doi.org/10.1016/j.neuron.2018.01.031
96. Hu, Y., Li, C., Wang, X., Chen, W., Qian, Y., & Dai, X. (2021). TREM2, Driving the Microglial Polarization, Has a TLR4 Sensitivity Profile After Subarachnoid Hemorrhage. Frontiers in cell and developmental biology, 9, 693342. https://doi.org/10.3389/fcell.2021.693342
97. Ulland, T. K., Song, W. M., Huang, S. C., Ulrich, J. D., Sergushichev, A., Beatty, W. L., Loboda, A. A., Zhou, Y., Cairns, N. J., Kambal, A., Loginicheva, E., Gilfillan, S., Cella, M., Virgin, H. W., Unanue, E. R., Wang, Y., Artyomov, M. N., Holtzman, D. M., & Colonna, M. (2017). TREM2 Maintains Microglial Metabolic Fitness in Alzheimer's Disease. Cell, 170(4), 649–663.e13.https://www.ncbi.nlm.nih.gov/pmc/articles/PMC5573224/
98. Spangenberg E, Severson PL, Hohsfield LA, Crapser J, Zhang J, Burton EA, Zhang Y, Spevak W, Lin J, Phan NY, Habets G, Rymar A, Tsang G, Walters J, Nespi M, Singh P, Broome S, Ibrahim P, Zhang C, Bollag G, West BL, Green KN. Sustained microglial depletion with CSF1R inhibitor impairs parenchymal plaque development in an Alzheimer's disease model. Nat Commun. 2019 Aug 21;10(1):3758. http://doi.org/10.1038/s41467-019-11674-z
99. Heneka, M. T., Kummer, M. P., Stutz, A., Delekate, A., Schwartz, S., Vieira-Saecker, A., Griep, A., Axt, D., Remus, A., Tzeng, T. C., Gelpi, E., Halle, A., Korte, M., Latz, E., & Golenbock, D. T. (2013). NLRP3 is activated in Alzheimer's disease and contributes to pathology in APP/PS1 mice. Nature, 493(7434), 674–678. https://doi.org/10.1038/nature11729
100. Lovászi, M., Branco Haas, C., Antonioli, L., Pacher, P., & Haskó, G. (2021). The role of P2Y receptors in regulating immunity and metabolism. Biochemical pharmacology, 187, 114419. https://doi.org/10.1016/j.bcp.2021.114419
101. Puigdellívol, M., Milde, S., Vilalta, A., Cockram, T. O. J., Allendorf, D. H., Lee, J. Y., Dundee, J. M., Pampuščenko, K., Borutaite, V., Nuthall, H. N., Brelstaff, J. H., Spillantini, M. G., & Brown, G. C. (2021). The microglial P2Y6 receptor mediates neuronal loss and memory deficits in neurodegeneration. Cell reports, 37(13), 110148. https://doi.org/10.1016/j.celrep.2021.110148
102. Moore CS, Ase AR, Kinsara A, Rao VT, Michell-Robinson M, Leong SY, Butovsky O, Ludwin SK, Séguéla P, Bar-Or A, Antel JP. P2Y12 expression and function in alternatively activated human microglia. Neurol Neuroimmunol Neuroinflamm. 2015 Mar 19;2(2):e80. http://doi.org/10.1212/NXI.0000000000000080
103. Ngolab, J., Trinh, I., Rockenstein, E., Mante, M., Florio, J., Trejo, M., Masliah, D., Adame, A., Masliah, E., & Rissman, R. A. (2017). Brain-derived exosomes from dementia with Lewy bodies propagate α-synuclein pathology. Acta neuropathologica communications, 5(1), 46. https://doi.org/10.1186/s40478-017-0445-5
104. Mallach, A., Zielonka, M., van Lieshout, V., An, Y., Khoo, J. H., Vanheusden, M., Chen, W. T., Moechars, D., Arancibia-Carcamo, I. L., Fiers, M., & De Strooper, B. (2024). Microglia-astrocyte crosstalk in the amyloid plaque niche of an Alzheimer's disease mouse model, as revealed by spatial transcriptomics. Cell reports, 43(6), 114216. Advance online publication. https://doi.org/10.1016/j.celrep.2024.114216
105. Lee, M., McGeer, E. G., & McGeer, P. L. (2016). Sodium thiosulfate attenuates glial-mediated neuroinflammation in degenerative neurological diseases. Journal of neuroinflammation, 13, 32. https://doi.org/10.1186/s12974-016-0488-8
106. Keren-Shaul, H., Spinrad, A., Weiner, A., Matcovitch-Natan, O., Dvir-Szternfeld, R., Ulland, T. K., David, E., Baruch, K., Lara-Astaiso, D., Toth, B., Itzkovitz, S., Colonna, M., Schwartz, M., & Amit, I. (2017). A Unique Microglia Type Associated with Restricting Development of Alzheimer's Disease. Cell, 169(7), 1276–1290.e17. https://doi.org/10.1016/j.cell.2017.05.018
107. Wen, L., You, W., Wang, H., Meng, Y., Feng, J., & Yang, X. (2018). Polarization of Microglia to the M2 Phenotype in a Peroxisome Proliferator-Activated Receptor Gamma-Dependent Manner Attenuates Axonal Injury Induced by Traumatic Brain Injury in Mice. Journal of neurotrauma, 35(19), 2330–2340. https://doi.org/10.1089/neu.2017.5540

Reviewer 2 Report

Comments and Suggestions for Authors

The manuscript ‘Aged-Related Changes in Microglia and Neurodegenerative Diseases: Exploring the Connection’ by Borrajo Ana provides an overview of the role of microglia in CNS homeostasis and aging, highlighting key factors regulating their development and maintenance. The discussion on the primed phenotype of aging microglia and its implications for neuroinflammation and cognitive impairment is important. Additionally, the exploration of microglia-mediated synaptic phagocytosis sheds light on its importance for neuronal plasticity. However, there are weaknesses that should be addressed:

Abstract:

The first sentence of the abstract should be modified to reflect that, while microglial cells share properties with macrophages, they also possess distinct characteristics. It could be revised to (or something similar): ‘Microglial cells exhibit properties akin to macrophages, thereby enabling them to…’

Introduction:

·         Paragraph #1: The first paragraph should be revised to provide more detail on the distinctions between microglia and other brain immune cells, such as macrophages, without oversimplifying their relationship. Explaining the unique genetic profiles and their impact on microglial functions would enhance understanding of their specialized role in brain immunity. Additionally, microglia originate from separate progenitor cells in the yolk sac during embryonic development, contrasting with macrophages. Supporting information for these distinct origins can be found in studies such as: 1.Ginhoux, F., & Prinz, M. (2015). Origin of microglia: current concepts and past controversies. Cold Spring Harbor perspectives in biology, 7(8), a020537. 2. Kierdorf, K., & Prinz, M. (2013). Factors regulating microglia activation. Frontiers in cellular neuroscience, 7, 44. 3. Gomez Perdiguero, E., Klapproth, K., Schulz, C., Busch, K., Azzoni, E., Crozet, L., Geissmann, F. (2015). Tissue-resident macrophages originate from yolk-sac-derived erythro-myeloid progenitors. Nature, 518(7540), 547-551.

·         Paragraph #2: The second paragraph of the introduction highlights the importance of CSF1, CSF1R, IL-34, IRF8, PU.1, and TGF-β in microglial function and their role in maintaining brain homeostasis, particularly in the context of neurodegenerative diseases like Alzheimer's and Parkinson's. However, it would be advantageous for the authors to acknowledge that while these factors are significant, there are likely numerous other factors and pathways involved in both brain homeostasis and disease pathology. For example, fractalkine (CX3CL1) and its sole receptor CX3CR1 signaling play a crucial role in microglia and neuron regulation. These additional factors may interact with or operate independently from the ones mentioned, contributing to the complex nature of neurodegenerative diseases.

·       Characteristics and Phenotypes of Aged Microglia:

·         Overall, the discussion about changes in aging microglia needs more context to understand their significance in brain aging, neuroinflammation, and neurodegenerative diseases.

·         Paragraph #1: ‘Higher microglia densities are exhibited compared to adjacent brain regions by the nigrostriatal system’ should be rephrased. 

·         In paragraph #2, organizing the content by species or by specific brain regions affected by aging could make it clearer and help compare studies. It briefly mentions contradictory findings, like how microglia numbers increase in aged rhesus monkeys but decrease in rats. Including these contradictions and talking about why they might happen would make the text more thorough. Also, giving context for each species or brain region discussed would help readers understand why the findings matter.

·         Paragraph 5: What are the specific implications of the altered microglial polarization for PD?

·         The paragraphs discussing age-related changes in microglial Ca2+ signaling repeatedly mention dysregulation of Ca2+ channels without providing a clear explanation of its significance or how it relates to the broader context of neurodegenerative diseases. By incorporating specific examples and structuring the text more cohesively, the discussion on age-related changes in microglial Ca2+ signaling can be made clearer and more informative for readers.

Aged-microglia and neurodegeneration:

The discussion doesn't connect the findings to specific neurodegenerative diseases or consider how diseases might vary. Also, it briefly talks about how problems with microglia could be important for treatments, but it doesn't offer specific treatment ideas or what future research could focus on.

Metabolism and oxidative stress in aged microglia:

The discussion doesn't clearly link the metabolic changes in aged microglia to specific neurodegenerative diseases or conditions. Additionally, while it mentions the potential therapeutic benefits of targeting certain metabolic pathways, it doesn't explore how feasible or effective these interventions would be in clinical practice.

Studies of aged microglia in Alzheimer’s and Parkinson’s disease:

The text explains how complicated microglial functions are in AD and PD, indicating that we still have a lot to learn about their role. It suggests discussing how this complexity makes it difficult to target microglia for treatments and offering possible solutions to this challenge.

Conclusion:

The conclusion needs more examples to support its points about aging and microglia. It should talk more about how we can treat microglial issues in diseases like Alzheimer's and Parkinson's. Also, it should mention any gaps in our understanding of microglial aging and diseases and give clearer ideas about what future research could focus on.

The authors should elaborate on the content presented in Figures 1 and 2, providing detailed explanations for better understanding in Figure Legend.

Author Response

The manuscript ‘Aged-Related Changes in Microglia and Neurodegenerative Diseases: Exploring the Connection’ by Borrajo Ana provides an overview of the role of microglia in CNS homeostasis and aging, highlighting key factors regulating their development and maintenance. The discussion on the primed phenotype of aging microglia and its implications for neuroinflammation and cognitive impairment is important. Additionally, the exploration of microglia-mediated synaptic phagocytosis sheds light on its importance for neuronal plasticity. However, there are weaknesses that should be addressed:
- Thank you for your very careful review of my paper, and for the comments, corrections and suggestions that ensued. I believe the paper has been significantly improved. In the final version of the manuscript, I have done the changes that you have suggested.

Abstract:
The first sentence of the abstract should be modified to reflect that, while microglial cells share properties with macrophages, they also possess distinct characteristics. It could be revised to (or something similar): ‘Microglial cells exhibit properties akin to macrophages, thereby enabling them to…’
According with the clever Reviewer’s comment, I have modified the text suggested. The new text appears in blue in the final version of manuscript.

Introduction:
   Paragraph #1: The first paragraph should be revised to provide more detail on the distinctions between microglia and other brain immune cells, such as macrophages, without oversimplifying their relationship. Explaining the unique genetic profiles and their impact on microglial functions would enhance understanding of their specialized role in brain immunity. Additionally, microglia originate from separate progenitor cells in the yolk sac during embryonic development, contrasting with macrophages. Supporting information for these distinct origins can be found in studies such as: 1.Ginhoux, F., & Prinz, M. (2015). Origin of microglia: current concepts and past controversies. Cold Spring Harbor perspectives in biology, 7(8), a020537. 2. Kierdorf, K., & Prinz, M. (2013). Factors regulating microglia activation. Frontiers in cellular neuroscience, 7, 44. 3. Gomez Perdiguero, E., Klapproth, K., Schulz, C., Busch, K., Azzoni, E., Crozet, L., Geissmann, F. (2015). Tissue-resident macrophages originate from yolk-sac-derived erythro-myeloid progenitors. Nature, 518(7540), 547-551.
-As you proposed, I have added the sentences that show the distinctions between microglia and other brain immune cells,
The text added in blue in the manuscript (1st paragraph) and the references (in blue):
Microglia originate from separate progenitor cells in the yolk sac during embryonic development, contrasting with macrophages. [4, 5, 6, 7]
“Also, microglia are located in the parenchyma and CNS-associated macrophages (CAMs) are found in CNS interfaces including the meninges, perivascular space, and choroid plexus. While microglia are the only immune cells located in the CNS parenchyma in close vicinity to neurons, the CAMs and additional immune cells, such as T and B cells, dendritic cells (DCs), monocytes, natural killer (NK) cells, and NKT cells, are found at the CNS borders, such as the meninges (leptomeninges, dura), and in the choroid plexus [6, 7].”
4. Prinz, M., & Priller, J. (2017). The role of peripheral immune cells in the CNS in steady state and disease. Nature neuroscience, 20(2), 136–144. https://doi.org/10.1038/nn.4475
5. Ginhoux, F., & Prinz, M. (2015). Origin of microglia: current concepts and past controversies. Cold Spring Harbor perspectives in biology, 7(8), a020537. https://doi.org/10.1101/cshperspect.a020537
6. Gomez Perdiguero, E., Klapproth, K., Schulz, C., Busch, K., Azzoni, E., Crozet, L., Garner, H., Trouillet, C., de Bruijn, M. F., Geissmann, F., & Rodewald, H. R. (2015). Tissue-resident macrophages originate from yolk-sac-derived erythro-myeloid progenitors. Nature, 518(7540), 547–551. https://doi.org/10.1038/nature13989
7. Kierdorf, K., & Prinz, M. (2013). Factors regulating microglia activation. Frontiers in cellular neuroscience, 7, 44. https://doi.org/10.3389/fncel.2013.00044
I have followed the Reviewer’s recommendation and have added the information about microglial functions. Text added and the references (appear in red in the final version of the paper):
“The use of colony-stimulating factor-1 receptor (CSF1R) inhibitors has provided significant insights into microglial dynamics in the adult CNS. In a interesant work, Elmore et al. in 2014 [12] observed that administering CSF1R inhibitors almost completely eliminated microglial cells in the adult CNS. Following the withdrawal of these inhibitors, microglia rapidly repopulated the CNS. This repopulation was associated with an increase in nestin-positive cells throughout the CNS, suggesting these cells represent microglial progenitors [12]. Initially, it was proposed that the repopulating microglia originated from a neuroectodermal lineage. This hypothesis was based on the observation that microglial cells are myeloid lineage cells and nestin-positive progenitor cells originate from the neuroectodermal lineage [13] suggested that repopulated microglia are derived from myeloid progenitor cells. However, recent findings challenge these theories. In a recent study, an adult mouse model have been used and it has been demonstrated that repopulated microglia do not originate from blood cells, nestin-positive cells, astrocytes, oligodendrocyte precursor cells, or neurons. Instead, after the selective elimination of more than 99% of microglia, the remaining microglia (< 1%) proliferated to repopulate the entire brain. Thus, the surviving microglia are the sole source of the new microglia, which rapidly repopulate the brain [13].
In aged mice, the cortex contains fewer microglia compared to young adult mice, and these cells are smaller, less symmetrical, more elongated, and have fewer branches [13]. Aging affects microglia through telomere shortening, DNA damage, and oxidative stress. Telomeres, which are the protective ends of chromosomes, shorten with age, correlating with a decline in microglial self-renewal [14]. Aged microglia express higher levels of tumor necrosis factor-α (TNF-α) and    interleukin IL-6, and their dystrophic morphology hinders spatial learning [13]. Senescent microglia show increased levels of pro-inflammatory cytokines, reduced levels of chemokines, and decreased phagocytosis of amyloid beta (Aβ) fibrils [13]. They also exhibit a 'primed' phenotype, characterized by an exaggerated and uncontrolled inflammatory response to immune stimuli [13].”
12. Elmore, M. R., Najafi, A. R., Koike, M. A., Dagher, N. N., Spangenberg, E. E., Rice, R. A., Kitazawa, M., Matusow, B., Nguyen, H., West, B. L., & Green, K. N. (2014). Colony-stimulating factor 1 receptor signaling is necessary for microglia viability, unmasking a microglia progenitor cell in the adult brain. Neuron, 82(2), 380–397. https://www.ncbi.nlm.nih.gov/pmc/articles/PMC4161285/
13. Xu Y, Jin MZ, Yang ZY, Jin WL. Microglia in neurodegenerative diseases. Neural Regen Res. 2021 Feb;16(2):270-280. http://doi.org/10.4103/1673-5374.290881
14. Wolf, S. A., Boddeke, H. W., & Kettenmann, H. (2017). Microglia in Physiology and Disease. Annual review of physiology, 79, 619–643. https://doi.org/10.1146/annurev-physiol-022516-034406

     Paragraph #2: The second paragraph of the introduction highlights the importance of CSF1, CSF1R, IL-34, IRF8, PU.1, and TGF-β in microglial function and their role in maintaining brain homeostasis, particularly in the context of neurodegenerative diseases like Alzheimer's and Parkinson's. However, it would be advantageous for the authors to acknowledge that while these factors are significant, there are likely numerous other factors and pathways involved in both brain homeostasis and disease pathology. For example, fractalkine (CX3CL1) and its sole receptor CX3CR1 signaling play a crucial role in microglia and neuron regulation. These additional factors may interact with or operate independently from the ones mentioned, contributing to the complex nature of neurodegenerative diseases.
- I have followed the intelligent advice of the Reviewer and have added text, in the successive  chapter 2 (page 4; 2º,3º and 4º paragraph in red) about these factors and others, such as, CD200, involved in brain homeostasis and pathology.
“In an interesting study, an immunophenotypic variation in brain microglial cells had been observed and suggested, noting that cortical and striatal microglia are similar, while hippocampal microglia display an profile between pro- and anti-inflammatory states [36]. Grabert et al., described how the microglial cells in the hippocampus decrease the expression of genes related to environment sensing, provoking more vulnerable to aging and disease related protein deposition [36]. It has also been observed that in healthy conditions, microglial cells in the hippocampus present a higher “immune-vigilant” phenotype. This can be related to a higher chronic inflammatory microglial response in AD pathology to plaque formation. Another particular features of the hippocampus that might influence microglial activity, are microglial cells density and the gray matter content [37]. The fractalkine receptor, CXCR3 has been shown to be involved in neuron-microglia communication, so that, hippocampal lower values of CXCR3 might provoke less microglial cells control [37]. These changes may influence microglial cells activity patterns and predispose this activity towards hippocampal-related neurocognitive diseases, where neuroinflammation and microglial cells activation and neuroinflammation play crucial roles [37]
   Temporal lobes including motor areas and the prefrontal cortex are extensively connected, playing important roles in top-down behavioral control and language processing exhibing the highest degree of age-related atrophy, in the case of prefrontal cortex, with a significant decline in prefrontal gray matter volume compared to other areas such as superior parietal cortices and the inferior temporal. In the healthy cortex [37]. Microglia/macrophage-specific inflammatory receptors CD11b expression on microglia is consistently lower than in the spinal cord, and the microglia protein, CD40,  expression is lower in the cortex compared to the cerebellum. Conversely, CXCR3 expression is higher in the cortex than in the cerebellum [37].
   Also, microglia constitute 12% of the cellular population in the substantia nigra, a particularly dense population [37]. CXCR3 is more abundant in the striatum than in other structures like the cerebellum [37]. Another important receptor in microglia-neuron interactions is the membrane glycoprotein CD200, which is downregulated with age in the substantia nigra [37]. A high density of microglial cells, reduced CD200 receptor expression, or a decreased dopaminergic population can diminish the regulatory influence of neurons on microglial cells. Such an environment may lead to a reduction in the production of trophic factors or an increase in inflammatory molecules, which can be detrimental over time.”
Also, additional references have been added and appear in red (as the new text) in the manuscript:
36. Grabert K., Michoel T., Karavolos M. H., Clohisey S., Baillie J. K., Stevens M. P., et al. (2016). Microglial brain region-dependent diversity and selective regional sensitivities to aging. Nat. Neurosci. 19 504–516. http://doi.org/10.1038/nn.4222
37. Bachiller S, Jiménez-Ferrer I, Paulus A, Yang Y, Swanberg M, Deierborg T, Boza-Serrano A. Microglia in Neurological Diseases: A Road Map to Brain-Disease Dependent-Inflammatory Response. Front Cell Neurosci. 2018 Dec 18;12:488. http://doi./10.3389/fncel.2018.00488

    Characteristics and Phenotypes of Aged Microglia:
 Overall, the discussion about changes in aging microglia needs more context to understand their significance in brain aging, neuroinflammation, and neurodegenerative diseases.

· Paragraph #1: ‘Higher microglia densities are exhibited compared to adjacent brain regions by the nigrostriatal system’ should be rephrased.  
- I have followed the Reviewer’s recommendation and have added the information about changes and different shapes and functions of microglia in neuroinflammation and neurodegenerative diseases. (Test added and references in red in the new version of the paper (page 3, 8º paragraph) and The phrase of the paragraph 1 has been rewritten.
“In the presence of pathology, microglial cells changes both their shape and function, undergoing a transformation from ramified to hypertrophic and ameboid phenotypes-a progression that continues to guide research today [31]. The build-up of senescent microglia, along with their impaired immune functions and interactions with other brain cells, could play a crucial role in the development of neurodegenerative diseases [31]. The unique characteristics of yolk sac-derived microglial cells, which have a limited capacity for repopulation, suggest an intrinsic link to their senescence, potentially contributing significantly to age-related neurodegenerative diseases [31]. 
Regarding the satellite microglia, this subgroup is now identified by its distinctive association with neuronal cell bodies [32]. The portion of the axon and the satellite microglial cells overlaps and action potentials are initiated [32]. It has been observed both during development and adulthood in mice, with a preferential association with excitatory neurons [32] and additionally, this subpopulation has been noted in the cerebral cortex of adult rats and adult non-human primates using non-invasive two-photon in vivo imaging [33], indicating conservation across species. These cells show soma migration, interacting with both Purkinje neuronal cell bodies and proximal dendrites, are less ramified than their cortical counterparts, and exhibit reduced surveillance. This finding suggests that satellite microglia are heterogeneous in their dynamics under steady-state conditions [33]. Beyond the distinct morphological properties of microglia revealed by light or fluorescence microscopy, high-resolution electron microscopy has uncovered the presence of microglia filled with cellular debris, similar to the fat granule or gitter cells initially described by Río-Hortega (1920), during aging, age-related sensory loss, and Werner syndrome in mice [34]. Recently, electron microscopy also identified a unique microglial population, the "dark" microglia, in the adult and aged mouse hippocampus (CA1 region and dentate gyrus), cerebral cortex, amygdala, and hypothalamus. These cells coexist with the typical microglia but display several ultrastructural features, including their size, morphology, long stretches of endoplasmic reticulum, interactions with neurons and synapses, and association with the extracellular space. Unlike typical microglia, they exhibit markers of oxidative stress, such as a condensed, electron-dense cytoplasm and nucleoplasm, giving them a dark appearance in EM, along with Golgi apparatus/endoplasmic reticulum dilation, mitochondrial alterations, and a partial to complete loss of heterochromatin pattern [34]. In neurons, changes to the heterochromatin pattern are associated with cellular stress, aging, and brain disorders such as schizophrenia and Alzheimer's disease [35].”
The reference added in red in the text:
31. Rim C, You MJ, Nahm M, Kwon MS. Emerging role of senescent microglia in brain aging-related neurodegenerative diseases. Transl Neurodegener. 2024 Feb 20;13(1):10. http://doi.org/10.1186/s40035-024-00402-3
32. Wogram, E., Wendt, S., Matyash, M., Pivneva, T., Draguhn, A., & Kettenmann, H. (2016). Satellite microglia show spontaneous electrical activity that is uncorrelated with activity of the attached neuron. European Journal of Neuroscience, 43(11), 1523-1534. https://doi.org/10.1111/ejn.13256
33. Stowell, R. D., Wong, E. L., Batchelor, H. N., Mendes, M. S., Lamantia, C. E., Whitelaw, B. S., & Majewska, A. K. (2018). Cerebellar microglia are dynamically unique and survey Purkinje neurons in vivo. Developmental neurobiology, 78(6), 627–644. https://doi.org/10.1002/dneu.22572
34. Hui, B., Zhang, L., Zhou, Q., & Hui, L. (2018). Pristimerin Inhibits LPS-Triggered Neurotoxicity in BV-2 Microglia Cells Through Modulating IRAK1/TRAF6/TAK1-Mediated NF-κB and AP-1 Signaling Pathways In Vitro. Neurotoxicity research, 33(2), 268–283. https://doi.org/10.1007/s12640-017-9837-3
35. Medrano-Fernández, A., & Barco, A. (2016). Nuclear organization and 3D chromatin architecture in cognition and neuropsychiatric disorders. Molecular brain, 9(1), 83. https://doi.org/10.1186/s13041-016-0263-x

·  In paragraph #2, organizing the content by species or by specific brain regions affected by aging could make it clearer and help compare studies. It briefly mentions contradictory findings, like how microglia numbers increase in aged rhesus monkeys but decrease in rats. Including these contradictions and talking about why they might happen would make the text more thorough. Also, giving context for each species or brain region discussed would help readers understand why the findings matter.
- Thank you for your very careful review of our paper. I have added this information in the text previously mentioned in Introduction paragraph 2. This text explain the specific brain regions affected by aging as requested by the reviewer.
The text added in red in the manuscript (page 4):
“In an interesting study, an immunophenotypic variation in brain microglial cells had been observed and suggested, noting that cortical and striatal microglia are similar, while hippocampal microglia display an profile between pro- and anti-inflammatory states [36]. Grabert et al., described how the microglial cells in the hippocampus decrease the expression of genes related to environment sensing, provoking more vulnerable to aging and disease related protein deposition [36]. It has also been observed that in healthy conditions, microglial cells in the hippocampus present a higher “immune-vigilant” phenotype. This can be related to a higher chronic inflammatory microglial response in AD pathology to plaque formation. Another particular features of the hippocampus that might influence microglial activity, are microglial cells density and the gray matter content [37]. The fractalkine receptor, CXCR3 has been shown to be involved in neuron-microglia communication, so that, hippocampal lower values of CXCR3 might provoke less microglial cells control [37]. These changes may influence microglial cells activity patterns and predispose this activity towards hippocampal-related neurocognitive diseases, where neuroinflammation and microglial cells activation and neuroinflammation play crucial roles [37]
Temporal lobes including motor areas and the prefrontal cortex are extensively connected, playing important roles in top-down behavioral control and language processing exhibing the highest degree of age-related atrophy, in the case of prefrontal cortex, with a significant decline in prefrontal gray matter volume compared to other areas such as superior parietal cortices and the inferior temporal. In the healthy cortex [37]. Microglia/macrophage-specific inflammatory receptors CD11b expression on microglia is consistently lower than in the spinal cord, and the microglia protein, CD40,  expression is lower in the cortex compared to the cerebellum. Conversely, CXCR3 expression is higher in the cortex than in the cerebellum [37].
Also, microglia constitute 12% of the cellular population in the substantia nigra, a particularly dense population [37]. CXCR3 is more abundant in the striatum than in other structures like the cerebellum [37]. Another important receptor in microglia-neuron interactions is the membrane glycoprotein CD200, which is downregulated with age in the substantia nigra [37]. A high density of microglial cells, reduced CD200 receptor expression, or a decreased dopaminergic population can diminish the regulatory influence of neurons on microglial cells. Such an environment may lead to a reduction in the production of trophic factors or an increase in inflammatory molecules, which can be detrimental over time.”
Also, additional references have been added and appear in red (as the new text) in the manuscript:
36. Grabert K., Michoel T., Karavolos M. H., Clohisey S., Baillie J. K., Stevens M. P., et al. (2016). Microglial brain region-dependent diversity and selective regional sensitivities to aging. Nat. Neurosci. 19 504–516. http://doi.org/10.1038/nn.4222
37. Bachiller S, Jiménez-Ferrer I, Paulus A, Yang Y, Swanberg M, Deierborg T, Boza-Serrano A. Microglia in Neurological Diseases: A Road Map to Brain-Disease Dependent-Inflammatory Response. Front Cell Neurosci. 2018 Dec 18;12:488. http://doi./10.3389/fncel.2018.00488

  Paragraph 5: What are the specific implications of the altered microglial polarization for PD?
- In accordance with Reviewer’s question, I have redacted text about specific marker as NF-κB and others (mentioned in the text) of the altered microglia.
“Also, NF-κB promotes the M1 microglia phenotype; on the other hand, there are a number of regulators that can inhibit NF-κB to switch microglia from M1 to M2. This indicates that NF-κB could be an important factor in microglia polarization. When in the cytoplasm, it is present in its inactive form without being stimulated to bind to IκB/NF-κB [47].
The idea of microglia polarization is currently debated because the M1/M2 paradigm may oversimplify in vivo activation PD. The heterogeneity of the microglia phenotype related to PD pathogenic situations can be further defined using transcriptomic and proteomic analyses”

·         The paragraphs discussing age-related changes in microglial Ca2+ signaling repeatedly mention dysregulation of Ca2+ channels without providing a clear explanation of its significance or how it relates to the broader context of neurodegenerative diseases. By incorporating specific examples and structuring the text more cohesively, the discussion on age-related changes in microglial Ca2+ signaling can be made clearer and more informative for readers.

- In accordance with Reviewer’s advice, I have added the text about dysregulation of CA2+
Text added in the manuscript and the reference (appear in blue in the final version of the manuscript (page 7, last paragraph) 
Generally, defects in neuronal autophagy regulation, related to Ca2+regulation in the ER and mitochondria, contribute to several diseases, such as AD and PD [56].
Early Ca2+ dysregulations are seen in both neurons and glial cells. In AD, deficits in microglial Ca2+ provoke signaling increase Aβ deposits and disrupt gliotransmitter release, which is necessary for good astrocyte–neuron communication in the CNS. If Ca2+dysregulation occurs first in glial cells and then propagates to neurons, targeting glia cells early in AD is a possible therapeutic target option [56]. Intracellular Ca2+homeostasis mediated by several organelles such as lysosomes, ER, and mitochondria, establish a mechanism for aggregated protein clearance and cellular stress regulation [56]. Any intracellular Ca2+dyshomeostasis contributes to proteinopathy, neuroinflammation, and ROS production, leadingto neurodegeneration [56].
56. Mustaly-Kalimi, S., Littlefield, A. M., & Stutzmann, G. E. (2018). Calcium Signaling Deficits in Glia and Autophagic Pathways Contributing to Neurodegenerative Disease. Antioxidants & redox signaling, 29(12), 1158–1175. https://doi.org/10.1089/ars.2017.7266

Aged-microglia and neurodegeneration:

The discussion doesn't connect the findings to specific neurodegenerative diseases or consider how diseases might vary. Also, it briefly talks about how problems with microglia could be important for treatments, but it doesn't offer specific treatment ideas or what future research could focus on.

Studies of aged microglia in Alzheimer’s and Parkinson’s disease:

The text explains how complicated microglial functions are in AD and PD, indicating that we still have a lot to learn about their role. It suggests discussing how this complexity makes it difficult to target microglia for treatments and offering possible solutions to this challenge.
-Thank you for your very careful review of our paper, and for the comments, corrections and suggestions that ensued. We believe the paper has been significantly improved. In the final version of the manuscript, I have written the additional chapter that answers the questions in these two sections:
7. Therapeutic targets for regulating microglial cells in Alzheimer and Parkinson diseases
(text and referenced added in red in the final version of the paper) (page 12).
86. Hansen, J. N., Brückner, M., Pietrowski, M. J., Jikeli, J. F., Plescher, M., Beckert, H., Schnaars, M., Fülle, L., Reitmeier, K., Langmann, T., Förster, I., Boche, D., Petzold, G. C., & Halle, A. (2022). MotiQ: an open-source toolbox to quantify the cell motility and morphology of microglia. Molecular biology of the cell, 33(11), ar99. https://doi.org/10.1091/mbc.E21-11-0585
87. Plescher, M., Seifert, G., Hansen, J. N., Bedner, P., Steinhäuser, C., & Halle, A. (2018). Plaque-dependent morphological and electrophysiological heterogeneity of microglia in an Alzheimer's disease mouse model. Glia, 66(7), 1464–1480. https://doi.org/10.1002/glia.23318
88. Yao, X., Liu, S., Ding, W., Yue, P., Jiang, Q., Zhao, M., Hu, F., & Zhang, H. (2017). TLR4 signal ablation attenuated neurological deficits by regulating microglial M1/M2 phenotype after traumatic brain injury in mice. Journal of neuroimmunology, 310, 38–45. https://doi.org/10.1016/j.jneuroim.2017.06.006
89. Liu, J. T., Wu, S. X., Zhang, H., & Kuang, F. (2018). Inhibition of MyD88 Signaling Skews Microglia/Macrophage Polarization and Attenuates Neuronal Apoptosis in the Hippocampus After Status Epilepticus in Mice. Neurotherapeutics : the journal of the American Society for Experimental NeuroTherapeutics, 15(4), 1093–1111. https://doi.org/10.1007/s13311-018-0653-0
90. Yang, Z., Liu, B., Zhong, L., Shen, H., Lin, C., Lin, L., Zhang, N., & Yuan, B. (2015). Toll-like receptor-4-mediated autophagy contributes to microglial activation and inflammatory injury in mouse models of intracerebral haemorrhage. Neuropathology and applied neurobiology, 41(4), e95–e106. https://doi.org/10.1111/nan.12177
91. Jain, M., Singh, M. K., Shyam, H., Mishra, A., Kumar, S., Kumar, A., & Kushwaha, J. (2021). Role of JAK/STAT in the Neuroinflammation and its Association with Neurological Disorders. Annals of neurosciences, 28(3-4), 191–200. https://doi.org/10.1177/09727531211070532
92. Butturini, E., Boriero, D., Carcereri de Prati, A., & Mariotto, S. (2019). STAT1 drives M1 microglia activation and neuroinflammation under hypoxia. Archives of biochemistry and biophysics, 669, 22–30. https://doi.org/10.1016/j.abb.2019.05.011
93. Pawelec, P., Ziemka-Nalecz, M., Sypecka, J., & Zalewska, T. (2020). The Impact of the CX3CL1/CX3CR1 Axis in Neurological Disorders. Cells, 9(10), 2277. https://doi.org/10.3390/cells9102277
94. Bolós, M., Llorens-Martín, M., Perea, J. R., Jurado-Arjona, J., Rábano, A., Hernández, F., & Avila, J. (2017). Absence of CX3CR1 impairs the internalization of Tau by microglia. Molecular neurodegeneration, 12(1), 59. https://doi.org/10.1186/s13024-017-0200-1
95. Zhao, Y., Wu, X., Li, X., Jiang, L. L., Gui, X., Liu, Y., Sun, Y., Zhu, B., Piña-Crespo, J. C., Zhang, M., Zhang, N., Chen, X., Bu, G., An, Z., Huang, T. Y., & Xu, H. (2018). TREM2 Is a Receptor for β-Amyloid that Mediates Microglial Function. Neuron, 97(5), 1023–1031.e7. https://doi.org/10.1016/j.neuron.2018.01.031
96. Hu, Y., Li, C., Wang, X., Chen, W., Qian, Y., & Dai, X. (2021). TREM2, Driving the Microglial Polarization, Has a TLR4 Sensitivity Profile After Subarachnoid Hemorrhage. Frontiers in cell and developmental biology, 9, 693342. https://doi.org/10.3389/fcell.2021.693342
97. Ulland, T. K., Song, W. M., Huang, S. C., Ulrich, J. D., Sergushichev, A., Beatty, W. L., Loboda, A. A., Zhou, Y., Cairns, N. J., Kambal, A., Loginicheva, E., Gilfillan, S., Cella, M., Virgin, H. W., Unanue, E. R., Wang, Y., Artyomov, M. N., Holtzman, D. M., & Colonna, M. (2017). TREM2 Maintains Microglial Metabolic Fitness in Alzheimer's Disease. Cell, 170(4), 649–663.e13.https://www.ncbi.nlm.nih.gov/pmc/articles/PMC5573224/
98. Spangenberg E, Severson PL, Hohsfield LA, Crapser J, Zhang J, Burton EA, Zhang Y, Spevak W, Lin J, Phan NY, Habets G, Rymar A, Tsang G, Walters J, Nespi M, Singh P, Broome S, Ibrahim P, Zhang C, Bollag G, West BL, Green KN. Sustained microglial depletion with CSF1R inhibitor impairs parenchymal plaque development in an Alzheimer's disease model. Nat Commun. 2019 Aug 21;10(1):3758. http://doi.org/10.1038/s41467-019-11674-z
99. Heneka, M. T., Kummer, M. P., Stutz, A., Delekate, A., Schwartz, S., Vieira-Saecker, A., Griep, A., Axt, D., Remus, A., Tzeng, T. C., Gelpi, E., Halle, A., Korte, M., Latz, E., & Golenbock, D. T. (2013). NLRP3 is activated in Alzheimer's disease and contributes to pathology in APP/PS1 mice. Nature, 493(7434), 674–678. https://doi.org/10.1038/nature11729
100. Lovászi, M., Branco Haas, C., Antonioli, L., Pacher, P., & Haskó, G. (2021). The role of P2Y receptors in regulating immunity and metabolism. Biochemical pharmacology, 187, 114419. https://doi.org/10.1016/j.bcp.2021.114419
101. Puigdellívol, M., Milde, S., Vilalta, A., Cockram, T. O. J., Allendorf, D. H., Lee, J. Y., Dundee, J. M., Pampuščenko, K., Borutaite, V., Nuthall, H. N., Brelstaff, J. H., Spillantini, M. G., & Brown, G. C. (2021). The microglial P2Y6 receptor mediates neuronal loss and memory deficits in neurodegeneration. Cell reports, 37(13), 110148. https://doi.org/10.1016/j.celrep.2021.110148
102. Moore CS, Ase AR, Kinsara A, Rao VT, Michell-Robinson M, Leong SY, Butovsky O, Ludwin SK, Séguéla P, Bar-Or A, Antel JP. P2Y12 expression and function in alternatively activated human microglia. Neurol Neuroimmunol Neuroinflamm. 2015 Mar 19;2(2):e80. http://doi.org/10.1212/NXI.0000000000000080
103. Ngolab, J., Trinh, I., Rockenstein, E., Mante, M., Florio, J., Trejo, M., Masliah, D., Adame, A., Masliah, E., & Rissman, R. A. (2017). Brain-derived exosomes from dementia with Lewy bodies propagate α-synuclein pathology. Acta neuropathologica communications, 5(1), 46. https://doi.org/10.1186/s40478-017-0445-5
104. Mallach, A., Zielonka, M., van Lieshout, V., An, Y., Khoo, J. H., Vanheusden, M., Chen, W. T., Moechars, D., Arancibia-Carcamo, I. L., Fiers, M., & De Strooper, B. (2024). Microglia-astrocyte crosstalk in the amyloid plaque niche of an Alzheimer's disease mouse model, as revealed by spatial transcriptomics. Cell reports, 43(6), 114216. Advance online publication. https://doi.org/10.1016/j.celrep.2024.114216
105. Lee, M., McGeer, E. G., & McGeer, P. L. (2016). Sodium thiosulfate attenuates glial-mediated neuroinflammation in degenerative neurological diseases. Journal of neuroinflammation, 13, 32. https://doi.org/10.1186/s12974-016-0488-8
106. Keren-Shaul, H., Spinrad, A., Weiner, A., Matcovitch-Natan, O., Dvir-Szternfeld, R., Ulland, T. K., David, E., Baruch, K., Lara-Astaiso, D., Toth, B., Itzkovitz, S., Colonna, M., Schwartz, M., & Amit, I. (2017). A Unique Microglia Type Associated with Restricting Development of Alzheimer's Disease. Cell, 169(7), 1276–1290.e17. https://doi.org/10.1016/j.cell.2017.05.018
107. Wen, L., You, W., Wang, H., Meng, Y., Feng, J., & Yang, X. (2018). Polarization of Microglia to the M2 Phenotype in a Peroxisome Proliferator-Activated Receptor Gamma-Dependent Manner Attenuates Axonal Injury Induced by Traumatic Brain Injury in Mice. Journal of neurotrauma, 35(19), 2330–2340. https://doi.org/10.1089/neu.2017.5540

Metabolism and oxidative stress in aged microglia:

The discussion doesn't clearly link the metabolic changes in aged microglia to specific neurodegenerative diseases or conditions. Additionally, while it mentions the potential therapeutic benefits of targeting certain metabolic pathways, it doesn't explore how feasible or effective these interventions would be in clinical practice.
- According with the clever Reviewer’s comment, I have added new text in red (page 9, last paragraph) and the reference (609 in red) in the final version of paper, to link some metabolic changes (mTORC signaling) with some neurodegenerative diseases. 
“mTORC1 signaling has been also identified as a promising target for the treatment of major depressive disorder. Luo et al. explored whether the mTORC1 signaling pathway is involved in synapse loss in the hippocampus caused by chronic stress. After successfully establishing the chronic restraint stress model, significant changes in the mRNA levels of certain immediate early genes were observed, indicating neuronal activation and changes in protein synthesis [69]. There was a notable downregulation of glutamate receptors and postsynaptic density protein at both the mRNA and protein levels. Synaptic fractionation assays indicated that chronic stress resulted in synapse loss in both ventral and dorsal hippocampus. These effects were related to the mTORC1 signaling pathway, as protein synthesis and phosphorylation of downstream signaling targets were reduced following chronic stress. Moreover, intracerebroventricular infusion of rapamycin induced depression-like behaviors and inhibited the antidepressant effects of fluoxetine [69]. This study demonstrates that the mTORC1 signaling pathway plays a crucial role in mediating synapse loss provoked by chronic stress in diseases such as AD and PD and contributes to the behavioral effects of antidepressant treatment.”
69. Luo YF, Ye XX, Fang YZ, Li MD, Xia ZX, Liu JM, Lin XS, Huang Z, Zhu XQ, Huang JJ, Tan DL, Zhang YF, Liu HP, Zhou J, Shen ZC. mTORC1 Signaling Pathway Mediates Chronic Stress-Induced Synapse Loss in the Hippocampus. Front Pharmacol. 2021 Dec 20;12:801234. http://doi./10.3389/fphar.2021.801234
Also, I have added a table (Table 1) that expresses the different advances on the therapeutic potential of microglia in diseases such as AD and PD.

Conclusion:

The conclusion needs more examples to support its points about aging and microglia. It should talk more about how we can treat microglial issues in diseases like Alzheimer's and Parkinson's. Also, it should mention any gaps in our understanding of microglial aging and diseases and give clearer ideas about what future research could focus on.
- I have followed the reviewer’s recommendation and I have tried to improve the conclusions, adding text (in blue) in the final version of the review.
“Recently, in the AD etiology literature, considerable attention has been paid to the emerging specialized functions of microglia in regulating synaptic development and degeneration, as synapse loss is the most directly relevant pathology to the development of cognitive deficits in AD. Many studies have repeatedly identified the microglia-synapse pathway as the crucial factor in AD pathogenesis. In future studies, how microglia respond to their environment and switch their phenotypes following the early stages of an insult should be determined in both the infant and aging brain, because current evidence shows that microglia with normal cellular homeostasis become aberrant and dysregulated during the aging process of the CNS, resulting in an increased susceptibility to subsequent immune challenges. In addition, genetic approaches should be applied to explore key targets contributing to the connections or activation of the microglia-synapse pathways to achieve early prevention and a potential cure for AD.”
“ Also, in different studies it have been found a connection between microglial activation and PD. α-synuclein might activate microglia through the TLR4 pathway, and that microglia, in turn, caused injury to dopaminergic neurons. Additionally, NLRP3 inflammasome signaling complex is involved in the induction of a pro-inflammatory state and α-synuclein activates NLRP3 inflammasome signaling in the microglia, and different α-synuclein species lead to distinct microglial responses via TLR ligation in PD models. Furthermore, the microglial cells with the accumulated α-synuclein exhibit phagocytosis and a high abundance of oxidative and pro-inflammatory species, provoking selective degradation of DA neurons and the recruitment of peripheral immune cells, promoting neuroinflammation and speed up the development of PD.”

The authors should elaborate on the content presented in Figures 1 and 2, providing detailed explanations for better understanding in Figure Legend.
- In accordance with intelligent Reviewer’s comment, I provide more specific explanations in the Figure Legend.
Figure 1. Schematic overview of the intrincate relationship between microglia and neuroinflammation. Microglia become activated (M1) due to ageing, oxidative stress, and different factors, which can lead to neuroinflammation and neurodegeneration. Activated microglia produce excessive ROS, ILs, TNF-α and others, triggering neuroinflammation to promote neuronal damage and cell death. M2 microglial phenotype contributes to the processes in phagocytosis and neuronal survival.

Figure 2. Diagram of the complement cascade. The complement can be activated by three separate pathways: the classical, lectin, and alternative pathways. There are three main complement activation pathways: the classical, lectin, and alternative. All of these routes lead to the cleavage of C3 and subsequently C5, leading to the opsonization of tissues, the production of C3a and C5a, and the assembly of the cytolytic membrane complex.

Round 2

Reviewer 2 Report

Comments and Suggestions for Authors

No comments.